# The emergence of circadian timekeeping in the intestine

Kathyani Parasram [1], Amy Zuccato[1], Minjeong Shin[2], Reegan Willms[2], Brian DeVeale[1], Edan Foley[2] & Phillip Karpowicz [1] ✉

The circadian clock is a molecular timekeeper, present from cyanobacteria to mammals, that coordinates internal physiology with the external environment. The clock has a 24-h period however development proceeds with its own timing, raising the question of how these interact. Using the intestine of *Drosophila melanogaster* as a model for organ development, we track how and when the circadian clock emerges in specific cell types. We find that the circadian clock begins abruptly in the adult intestine and gradually synchronizes to the environment after intestinal development is complete. This delayed start occurs because individual cells at earlier stages lack the complete circadian clock gene network. As the intestine develops, the circadian clock is first consolidated in intestinal stem cells with changes in Ecdysone and *Hnf4* signalling influencing the transcriptional activity of Clk/cyc to drive the expression of *tim*, *Pdp1*, and *vri*. In the mature intestine, stem cell lineage commitment transiently disrupts clock activity in differentiating progeny, mirroring early developmental clock-less transitions. Our data show that clock function and differentiation are incompatible and provide a paradigm for studying circadian clocks in development and stem cell lineages.

Circadian rhythms are 24-h cycles of physiological activity driven by the circadian clock, a molecular pacemaker found in nearly all cells of the body[1,2]. The circadian clock promotes health by coordinating tissues in the body to anticipate daily environmental changes[3]. In many animals, the circadian system is hierarchical, consisting of a central pacemaker in the brain and peripheral pacemakers located in other organs[4]. How the circadian clock develops is not well understood, but it arises during early embryogenesis in fish[5] and late fetal and postnatal development in humans[6]. The central pacemaker in mice arises earlier than clocks in the rest of the body[7–9] which suggests that certain tissue clocks are suppressed during development[10]. In *Drosophila melanogaster*, larval behaviors such as photo-avoidance show circadian rhythms[11–15] and the central pacemaker can be synchronized to the environment as early as first instar larva[16]. These observations raise questions about when and how the circadian clock emerges in specific tissues, and the genetic and/or cellular mechanisms that underlie the birth of the timing mechanism.

The circadian clock of *Drosophila* is a transcription-translation cycle in which the Circadian locomotor output cycles kaput or clock (Clk) / cycle (cyc) heterodimer activates gene expression while two of its targets, *period* (*per*)[17] and *timeless* (*tim*)[18], repress Clk/cyc activity to restart the cycle (Fig. 1A)[19]. A secondary loop consisting of *Par-domain protein 1* (*Pdp1*) and *vrille* (*vri*) stabilizes these rhythms by regulating *Clk*[1,20,21]. In *Drosophila*, peripheral clocks can be synchronized to the environment indirectly by the central pacemaker in the brain[1] and directly through the blue-light photoreceptor *cryptochrome* (*cry*)[22].

To address the question of how circadian clock activity is established during development, we tested a *Drosophila* tissue with a simple lineage[23,24], well-established developmental stages[25,26], and adult circadian clock activity[27,28]. The adult intestine is derived from a population of cells known as adult midgut precursors (AMPs) that proliferate throughout larval development[29–32] and pupation[30,33,34], until the intestine matures three days after emergence as an adult (eclosion)[29,31,35]. The intestine consists of intestinal stem cells (ISCs) which divide into

[1]Department of Biomedical Sciences, University of Windsor, Windsor, ON N9B 3P4, Canada. [2]Department of Medical Microbiology and Immunology, University of Alberta, Edmonton, AB T6G 2E1, Canada. ✉e-mail: phillip.karpowicz@uwindsor.ca

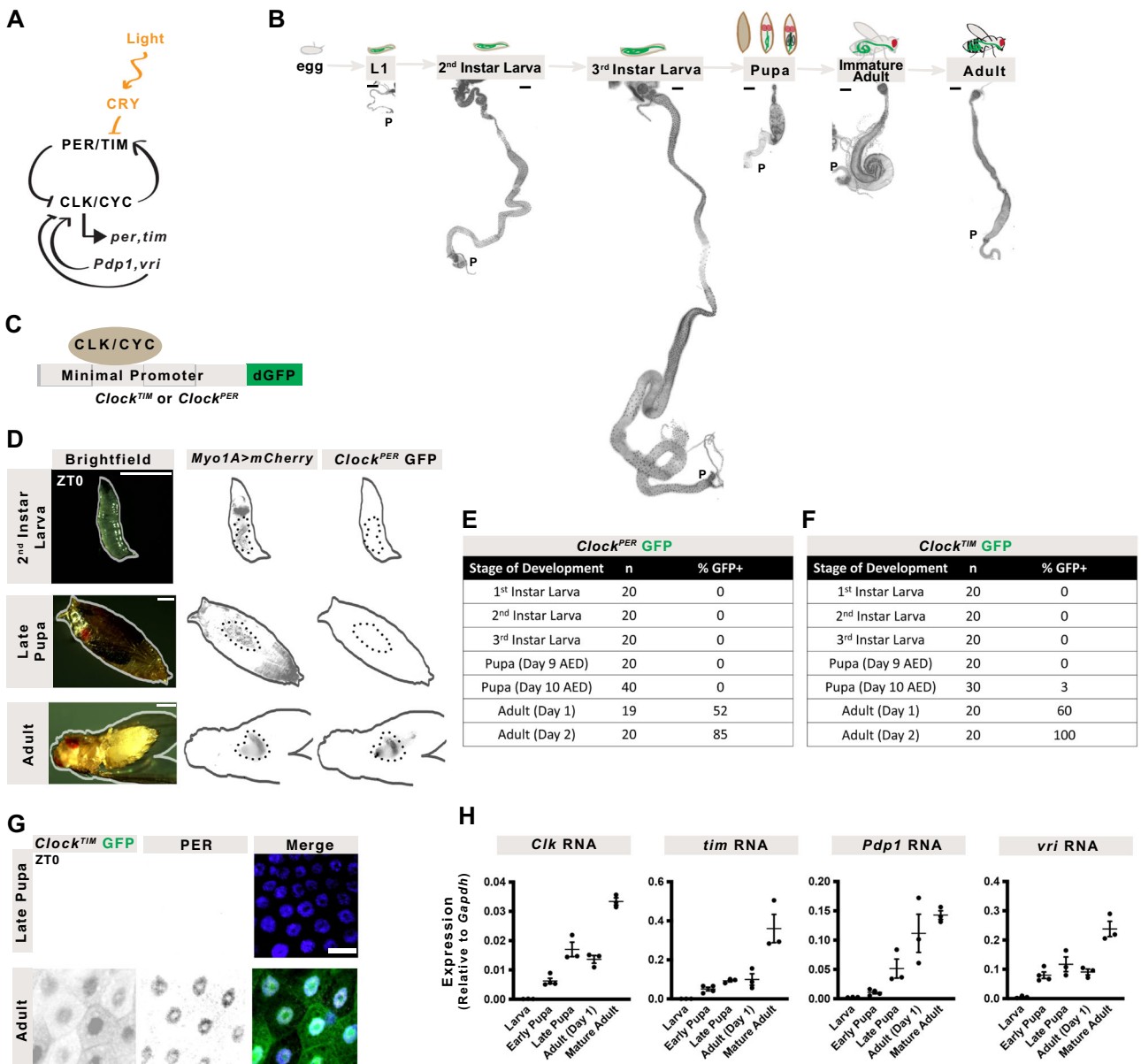

**Fig. 1 | The circadian clock emerges in the adult intestine. A** The circadian clock comprises the transcriptional activators Clk/cyc and their repressors per/tim. The repressor vri and activator Pdp1 form a secondary feedback system and light is sensed through cry. **B** *Drosophila* development from egg through three larval instars, followed by pupation prior to eclosion as an adult. Images of the midgut from larva to adults shows dramatic growth and remodeling of the intestine, DAPI stains nuclei (black). All images shown at same scale (scale bar = 200 μm). "P" indicates posterior. **C** Schematic of the Clock Reporter showing binding of Clk/cyc to the minimal promoter of either *tim* (for the *Clock^{TIM}* reporter) or *per* (*Clock^{PER}*) located upstream of destabilized GFP (dGFP). **D** Images of larva, pupa, and adult show that *Clock^{PER}* is expressed only in adult intestines (*Myo1A>mCherry*). The intestine is outlined in a dashed line (scale bar = 500 μm). Whole mount quantification of the number of flies expressing (**E**) *Clock^{PER}* or (**F**) *Clock^{TIM}* in the intestine throughout development. **G** Antibody staining showing that PER protein is only expressed in the adults and is localized to the nuclei in *Clock^{TIM}* flies at ZT0 (Zeitgeber Time relative to photoperiod, ZT0=lights on, ZT12=lights off). Scale 10 μm. Histone marks nuclei. **H** The mRNA expression of circadian clock genes at ZT0 from 3rd instar larva to adults showing the circadian clock transcripts increase in adults. One-Way ANOVA $p < 0.0001$ Clk and vri, $p = 0.0001$ tim and Pdp1. Error bars indicate ±SEM. Each point represents one replicate (20 intestines). Representative images of two replicates. Full statistics are shown in Supplementary Information. Related to Supplementary Figs. 2–3. Source data are provided as a Source Data file. L1 1st Instar Larva.

progenitor cells, known as enteroblasts (EBs) or enteroendocrine mother cells, that then differentiate into enterocytes (ECs) or enteroendocrine cells (EEs), respectively[23,24,36]. In the adult, the circadian clock controls ISC mitosis during regeneration[27], and we have previously shown that epithelial cells of the intestine have clock function[28].

Our results demonstrate that clock activity arises only after eclosion where it is synchronized to the environment by photoperiod and feeding over the first three days of adulthood. We characterize the transcriptome of single cells in the *Drosophila* intestine during adult maturation and clock formation and identify key regulators of this developmental transition. We propose a model where the ISCs are the first cells to have robust clock gene expression that is disrupted during cellular differentiation to be resumed in specialized tissue cells. We further suggest that complete clock gene networks are absent in EEs and EBs, which express only a subset of clock genes. Insects represent the most abundant and diverse species of animal on Earth, hence, our observations are significant in understanding how a critical physiological process is established in these animals and functions in their

adulthood. The maturation of the *Drosophila* intestinal clock is also reminiscent of mammalian / human development, where tissue clocks are synchronized to daily timing postnatally. Thus, our results establish a paradigm for the birth of circadian timekeeping in animal tissues by providing a transcriptomic framework of clock development at the single cell level.

## Results

### The circadian clock emerges in the adult intestine

During post-embryonic development the intestine undergoes significant growth and remodeling (Fig. 1B). We first determined if the circadian clock affects intestinal development by comparing isogenic wildtype and *per^01* mutant flies that exhibit no clock rhythms. Intestinal size, cell density, and the number of cells in AMP / PC islets do not differ significantly in *per^01* mutants (Supplementary Fig. 2A, B), indicating that intestinal development is not affected in the absence of circadian clock activity. To determine when Clk/cyc transcriptional activity begins, we used two different reporter constructs (*Clock^PER* and *Clock^TIM*) inserted on different chromosomes that use destabilized GFP (dGFP) to visualize Clk/cyc transcriptional activity on *per* and *tim* promoters with ~1 h temporal resolution[28] (Fig. 1C). The intestine was visualized using the GAL4/UAS system (*Myo1A>mCherry*) through the marker *Myosin 31DF* (*Myo1A*) that is expressed in both larval and adult intestinal cells[37]. No Clk/cyc transcriptional activity is present in larval (L1, L2, L3) or pupal flies, however, immediately after eclosion (adult day 1), Clk/cyc become active and remain so in later adulthood (Fig. 1D–F, Supplementary Fig. 2C, D) irrespective of sex or reproduction status (Supplementary Fig. 2E–G. To verify our reporter, we probed for per protein using both an antibody and a C-terminal GFP fusion reporter (*per-AID-eGFP*)[38], and cry protein using an N-terminal fusion protein for cry (*cry-GFP*)[39]. In all cases, these clock proteins are absent in pupa but present in recently eclosed adults (Fig. 1G and Supplementary Fig. 3A–C). We also verified that the clock is not present in larva and pupa, in a circadian phase opposite to the adults, by checking expression at different timepoints (Supplementary Fig. 3D). These results indicate that circadian clock activity begins only after pupal metamorphosis in the *Drosophila* intestine.

We next used RT-qPCR to examine the transcriptional expression of the core clock genes (*Clk, cyc, per, tim*), the secondary stabilizing loop (*Pdp1, vri*), the blue-light photoreceptor (*cry*), and *doubletime* (*dbt*), a kinase that regulates *per* to reset Clk/cyc driven transcription[40,41]. The intestinal expression of *Clk, tim, Pdp1E*, and *vri* increase throughout development to peak in adulthood, whereas *dbt, cyc, per*, and *cry* expression increase earlier in development before the stage when Clk/cyc activity exists (Fig. 1H and Supplementary Fig. 3E). Therefore, we hypothesized that the changes in clock activity are likely due to changes in *Clk, tim, Pdp1* and/or *vri*.

When we looked more closely at Clk/cyc activity in the last stages of pupation we noticed few scattered GFP cells (Supplementary Fig. 3F), raising the question of which intestinal cell type expresses Clk/cyc activity first. We measured Clk/cyc activity in pupal and adult ISC/EBs (marked by *esg>mCherry*) and ECs (marked by *Myo1A>mCherry*) and found that these early ISC/EBs show similar Clk/cyc activity in the late pupa and adults, unlike ECs that have significantly higher Clk/cyc activity in adults, suggesting that the ISC/EBs establish their circadian clock first (Supplementary Fig. 3G, H). Moreover, when the circadian clock is absent in all cells except ISC/EBs (*cyc^0* clock-dead mutant with *esg>cyc* rescue) clock activity persists in the ISC/EBs indicating that these cells can develop clocks independently of the other clocks in the body (Supplementary Fig. 3I).

### Daily clock rhythms are established after three days of adulthood

Since the intestinal clock emerges after pupation, we asked whether it is rhythmic immediately, with daily maxima and minima that are

characteristic of circadian rhythms, or whether it needs time to synchronize to the environment. We tested wildtype *CantonS* fly intestines by RT-qPCR to compare adult day 1 (immature) and day 4 (mature) gene expression patterns over a 24-h period. On day 1, the transcripts of *Pdp1* and *vri* are not rhythmic suggesting that the transcriptional cycle is not yet present. These genes increase in rhythmicity three days later, with *cry* changing the timing of its initial rhythms (Fig. 2A). Of note, the expression of all clock genes, with the exception of *cry*, are lower on day 1 than day 4, suggesting that establishment of robust transcriptional cycles takes several days.

The gradual emergence of transcriptional rhythmicity suggests Clk/cyc transactivation is not synchronized until the intestine matures. To test this, we imaged the *Clock^PER* and *Clock^TIM* reporters and measured fluorescence intensity in the whole intestine following pupation every 3-h over the first four days of adulthood (Fig. 2B, C and Supplementary Fig. 4A, B). When the adult ecloses, Clk/cyc transcriptional activity is present, but its rhythms are not robust 24-h circadian oscillations until day 4 when these consolidate to exhibit a maximum in the morning (ZT21-ZT0) and minimum in the evening (ZT9-ZT12) characteristic of the *Drosophila* intestine[28]. In each individual midgut, we did not notice major differences between cells which would indicate that the clocks in individual cells are at drastically different phases. To further test the synchrony between individual cells, we quantified EC fluorescence and noted that, although weaker rhythms are present at day 1, robust EC-specific rhythms are consolidated on day 4 (Supplementary Fig. 4C). This suggests Clk/cyc activity at both *per* (Fig. 2B) and *tim* (Fig. 2C) promoters synchronize over the first three days in the adult intestine.

A defining characteristic of circadian rhythms is their ability to continue in the absence of environmental cues such as photoperiod[42]. Accordingly, *Clock^TIM* intestines were examined when flies were shifted to constant darkness (DD) prior to ZT0 on day 4 and tracked for 48-h (Fig. 2D). The rhythms are similar to those under LD conditions, with a maximum in the morning (CT21-0) and minimum in the evening (CT12), demonstrating that Clk/cyc activity is free-running at day 4.

In a functioning circadian system, clock proteins shuttle between the nucleus and cytoplasm[43]. We tested a GFP reporter, *per-AID-eGFP*[38] at two timepoints ZT0 and ZT12 that represent nuclear and cytoplasmic localization of *per*, respectively[27]. Larva and pupa do not show any expression of per consistent with our qPCR and reporter analysis; the immature adult (day 1) shows signal at ZT0 that remains at ZT12, suggesting that the protein shuttling is not yet established, while the mature adult (day 4) shows strong nuclear signal at ZT0 but not at ZT12 (Fig. 2E). Taken together, our results show that circadian clock development in the intestine has three distinct phases: (1) embryogenesis to late pupa when the circadian clock circuit is absent, (2) between adult day 1 to 3 when Clk/cyc initiate clock rhythmicity but not yet fully synchronized to the environment, and (3) day 4 when the circadian clock is synchronized and rhythmic in the mature intestine.

### Clock gene expression increases in ISCs and ECs during intestinal maturation

Previous studies have not been able to examine circadian clock development in individual cells of a tissue, hence it is not clear if clock components mature homogenously, with individual cells gradually expressing all clock genes, or heterogeneously, with individual cells expressing certain clock genes but not others. To further test the development of the circadian pacemaker, we profiled transcriptional changes in *CantonS* intestinal development over each of the three stages in clock development, from early pupa, immature adults (day 1), and mature adults (day 7–8) using single-cell RNA sequencing (scRNA-seq). Early pupal intestines do not have Clk/cyc activity, immature adult intestines have Clk/cyc activity but no rhythms, and mature adult intestines have rhythmic Clk/cyc activity; we hypothesized that clock development would increase homogenously in all intestinal cell types

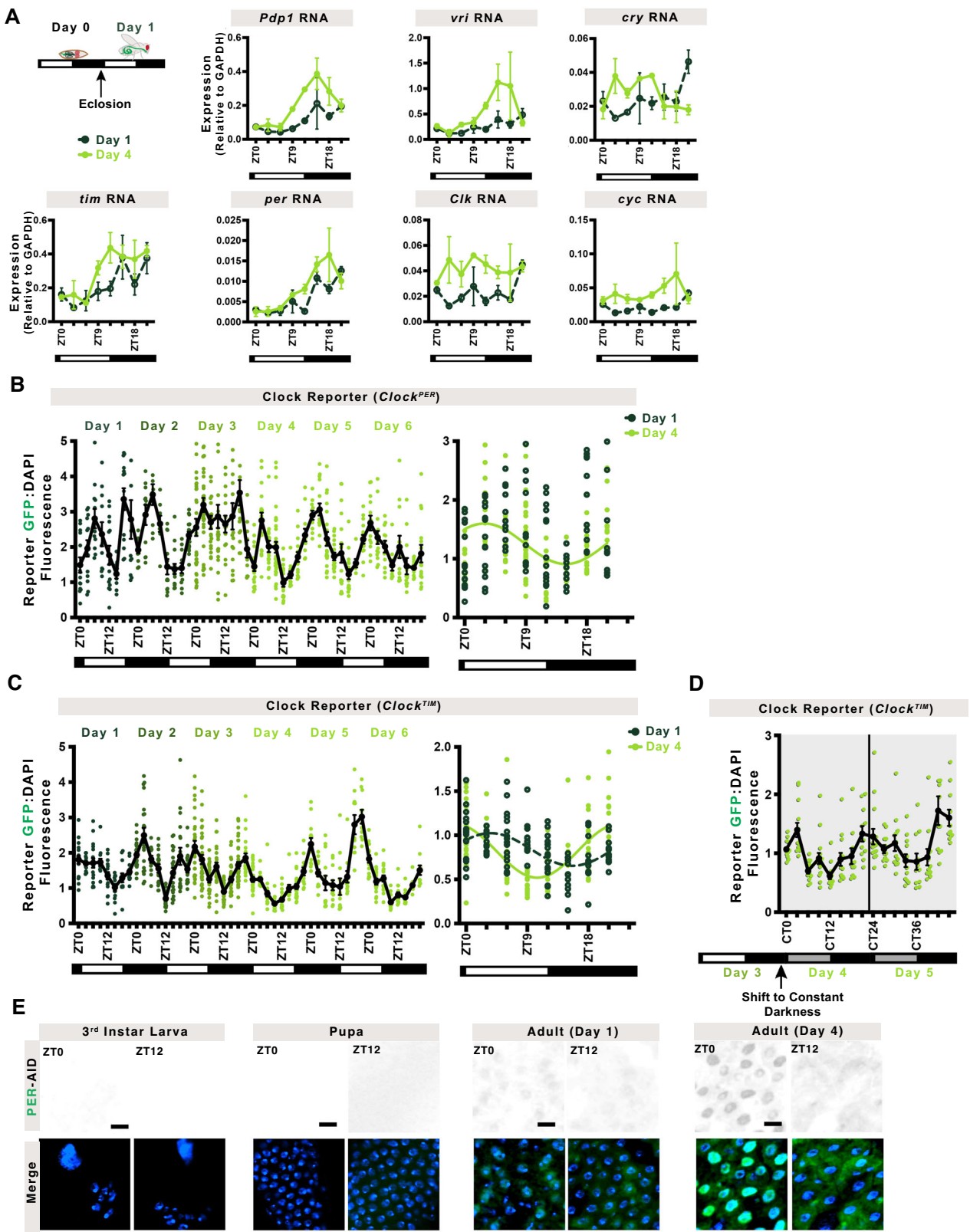

over these developmental stages. We recovered 5190 cells (2013 pupal, 1197 immature adult, and 1980 adult cells) that show 23 different cell states (Fig. 3A and Supplementary Fig. 5A, B), consistent with previous scRNA-seq studies of the *Drosophila* intestine[35,44–47], capturing the changing intestinal transcriptome throughout development. Our data provides a resource for cellular development in the insect intestine.

During the transition from pupa to mature adult, the cell population of the *Drosophila* intestine undergoes cellular changes (Fig. 3B). The ISCs and EEs are present at all stages but ECs are more restricted to later stages. A population of cells we denote as Pupal Cells (PupC) decrease in the immature adult and are completely absent in the mature adult. PupCs express some genes found in the EEs and ECs of

**Fig. 2 | Daily rhythms are established after three days of adulthood. A** Circadian clock genes are expressed in adults immediately after eclosion (day 1, shown in the schematic) but rhythms are stronger at day 4. For instance, amplitude of mRNA rhythm of *Pdp1* increases, and *cry* timing changes, other genes similarly show differences as intestine matures. Line shows mean, day 1 is shown in a dark green dashed line, day 4 in a light green solid line. Error bars indicate ±SEM. Each point is the average of two replicates each consisting of 15–20 intestines, *n* = 2. Two-Way ANOVA Pdp1 *p* = 0.3867; vri *p* = 0.2637; cry *p* = 0.0270; tim *p* = 0.5207; per *p* = 0.2634; Clk *p* = 0.5399; cyc *p* = 0.5007. **B** Clk/cyc transcriptional activity in the *Clock*[PER] reporter shows daily rhythms are established robustly on day 4 with a peak around ZT0 and trough around ZT12. Cosinor fit analysis (right graph) shows arrhythmic activity on Day 1 (no cosinor curve can be fitted to the data) and 24-h rhythms on Day 4 shown in a light green solid line. Lines show mean. Error bars indicate ±SEM. Each point represents one intestine, *n* = 737 intestines over at least two independent replicates. **C** In the *Clock*[TIM] reporter rhythms show a similar trend but are noted earlier, from day 1, that phase shift to match those of *Clock*[PER] by day 4. Lines show mean, day 1 dark green dashed line, day 4 light green solid line. Error bars indicate ±SEM. Each point represents one intestine, *n* = 853 intestines over at least two independent replicates. Day 1 corresponds to the first light and dark cue after eclosion. **D** Clk/cyc activity when shifted to constant darkness prior to ZT0 on Day 4 shows free-running rhythms over the first two days in constant darkness. The vertical line separates the first and second days in constant darkness. One-Way ANOVA p < 0.0001. Lines show mean. Error bars indicate ±SEM. Each point represents one intestine, *n* = 198 intestines over two independent replicates. **E** Protein expression of per (PER-AID-eGFP) at ZT0 and ZT12 shows that there is no per protein expression in larva or pupa, but per is expressed in day 1 or day 4 adults with clear nuclear (ZT0) and cytoplasmic (ZT12) staining established by day 4. Scale 10 μm. DAPI marks nuclei. Representative images of two replicates. Full statistics are shown in Supplementary Information. Related to Supplementary Figs. 3–4. Source data are provided as a Source Data file.

the adult (Supplementary Data 4)[45] and are likely derived from AMPs but represent a separate cell type only present in pupae (Supplementary Fig. 5C), consistent with previous reports that show transient pupal midgut cells disappear during metamorphosis[25]. Several sub-types of differentiated ECs are not present in the pupae or immature adult and are only present in the mature adult, consistent with the maturation and growth of the *Drosophila* intestine[48]. To assess circadian gene expression in these cells, we surveyed gene counts in individual cells (Supplementary Fig. 5D and Supplementary Data 1–3). We were able to detect all core clock components (Supplementary Fig. 5D) whose expression at the population level increases as pupa transition to mature adults consistent with the RT-qPCR data (Fig. 1H and Supplementary Fig. 3E). We detected the genes *per, Clk*, and *cyc* at much lower levels than *tim, Pdp1, vri, dbt*, and *cwo*, a repressor of Clk/cyc. We therefore used the higher-expressed clock genes as a readout of clock development.

Previously, we reported that ISCs, EBs, and ECs, but not EEs have Clk/cyc transcriptional rhythms[28]. We therefore asked whether individual cells, including EEs, during these developmental stages express certain clock genes, but not the complete clock system. To test this, we focused on clock target genes that are more highly expressed and also feedback to regulate the core clock (*cwo, tim, Pdp1* and *vri*). We graphed all four genes in a multi-dimensional fashion to simultaneously determine changing expression of all four in individual cells (*tim* is y-axis, *cwo* is x-axis, *vri* is size, *Pdp1* is color). Individual ISCs show low expression of all four genes in the pupa, increasing co-expression in the immature adults, that further increases in the mature adults (Fig. 3C). This suggests that the arrhythmic Clk/cyc activity on day 1 is, at least in part, due to low expression of the circadian clock program in individual ISCs. Individual ECs show very similar patterns of change, gradually single cells develop that express all four clock genes at higher levels. In contrast, clock genes are not expressed in individual EBs and EEs in the same manner. First, these do not show the robust expression of multiple clock genes at any developmental stage including the mature intestine. For example, only 10% of mature adult EBs coexpress 3–4 of the clock genes compared to 98% of mature adult ISCs. Second, in many cells one or two clock genes are high (Supplementary Fig. 5D), but the remainder are low, suggesting clock components may have non-clock functions in specific subtypes of EEs or EBs but are not co-expressed as part of a circadian expression program. The highest clock gene expression is present in ISCs, that increase expression in the immature and mature adult intestine, whereas the other intestinal cell types showed lower expression levels. Finally, individual PupCs do not express clock genes or express few at very low levels, which is consistent with the absence of Clk/cyc activity in the pupal intestine (Fig. 3D). We conclude that ISCs and ECs increase clock gene expression after pupation, supporting a model where clock gene expression that is absent earlier in development increases in

single cells homogenously to reach its maximum in the mature adult intestine.

## Hormonal regulators of clock emergence

The sudden emergence of Clk/cyc activity in the immature adult intestine (Fig. 1D–F) suggests that either a regulatory factor initiates the expression of circadian clock genes in the adult, or else suppresses them in the pupa, until the appropriate developmental stage is completed. Hormone nuclear receptors control many aspects of *Drosophila* physiology and metamorphosis[49,50] and are intimately connected with circadian clock function[51]. We therefore hypothesized that clock development is coordinated by hormone nuclear receptor signaling in the developing intestine. We used SCENIC[52], a bioinformatic approach, that evaluates scRNA-seq transcriptomes to identify active gene networks and infer the underlying transcriptional regulators, to analyze pupal, immature, and mature adult cells. Identification of cell-specific transcription factors revealed enrichment of *klu* and *Sox21a* in ISCs, known to regulate ISC differentiation/function[45,53–55], the Notch target *E(spl)* in ISC/EBs, known to drive ISC differentiation[56,57], *nub/Pdm1* in ECs, a well-known marker of this cell type[23,24], *tap* in EEs[58], known to promote EE cell fate[59] (Fig. 4A). Hence SCENIC is able to identify cell-specific transcriptional networks in the *Drosophila* intestine accurately.

To infer regulators of clock differentiation we focused on the ISC population, since it is present at all three stages in intestinal development (Fig. 3B and Supplementary Fig. 5A). We evaluated differences in regulon activity between the early pupa and immature adults to specifically address the changes occurring during these developmental stages as the clock first emerges (Fig. 4B and Supplementary Data 4). This analysis identified increased activity of 13 transcription factors, and decreased activity of 17 others during pupation. These include known ISC regulators *Stat92E*[60–64], and *GATA*[65], the hormone signaling factors *gce/Met* (germ cell-expressed bHLH-PAS/Methoprene-tolerant)[66,67], and *EcR* (ecdysone receptor)[68–70] that are active during metamorphosis[69,71,72], and the clock gene *Pdp1*. Many other factors were identified that are specific to different cell types in the developing intestine. Our dataset provides a resource for the global transcriptional changes that occur during the growth and development of the *Drosophila* ISC model system.

*Drosophila* pupation consists of sequential hormonal pulses with rhythms in adult emergence occurring through circadian control of the EcR[73]. We find that *EcR* and the nuclear receptor *ftz-f1* (fushi tarazu transcription factor 1), are downregulated in the immature adult ISCs, whereas *gce* and *Met*, both of which are involved in Juvenile hormone signaling during metamorphosis[66,67], are upregulated (Fig. 4B, C). Some of these differentially expressed regulons also show clear bimodal distributions indicating that there are distinct cell populations, with and without activity of these transcription factors (*i.e. gce,*

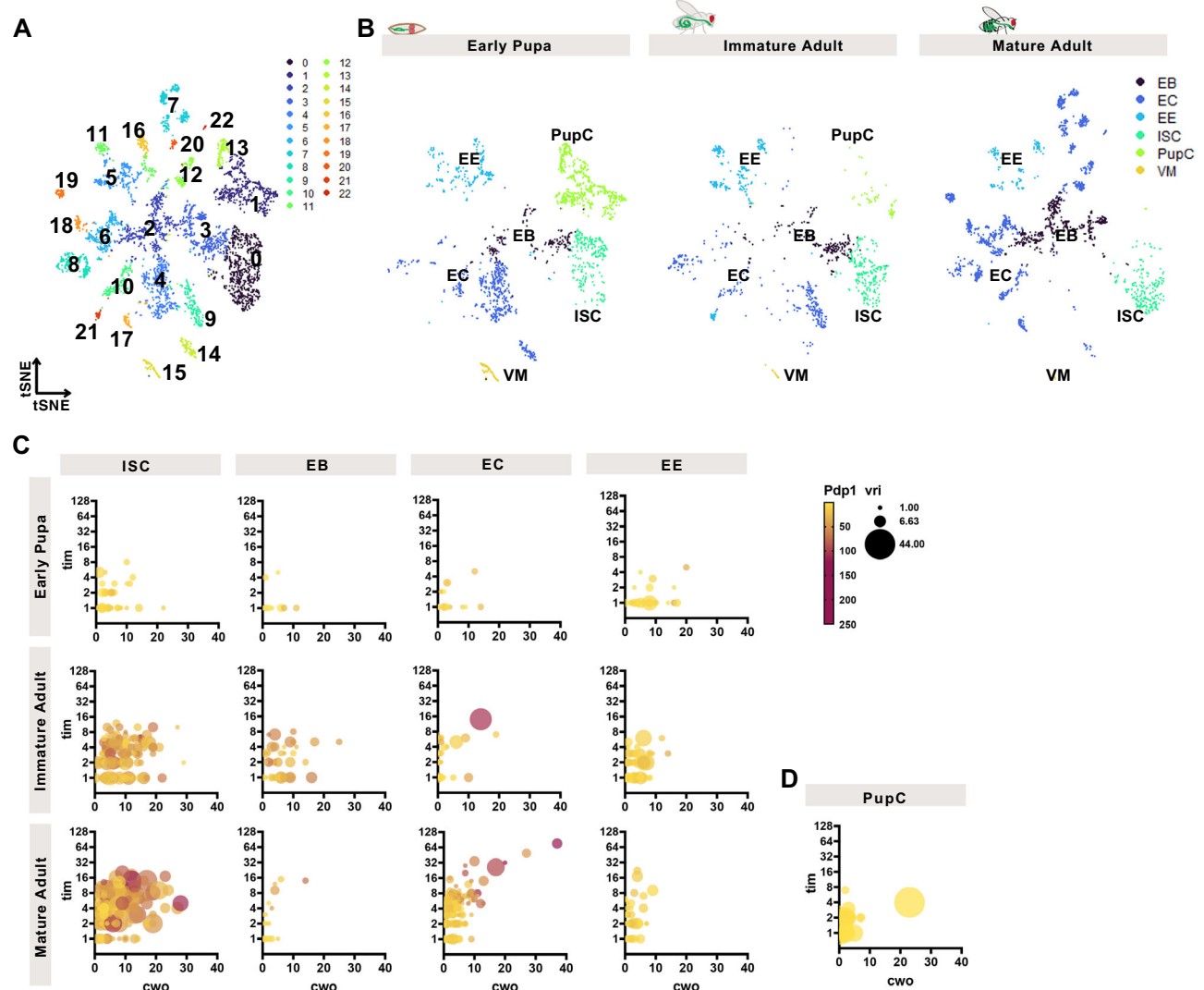

**Fig. 3 | Clock gene expression increases in ISCs and ECs during intestinal maturation.** scRNA-seq analysis of clock gene expression of the intestine of early pupa, day 1 immature adults, and day 7-8 mature adults showing (**A**) tSNE plots with 23 different cell populations over the integrated dataset. **B** Population changes are evident during three developmental stages with PupCs (green) disappearing from the pupal intestine, and ECs (dark blue) appearing in distinct clusters in the mature intestine. ISCs (aquamarine) are present at all stages, similar to EEs (light blue). For cluster assignments and markers see Supplementary Fig. 5A and Supplementary Data 4. **C** Multidimensional plots showing the changes in four highly expressed clock genes from early pupa to adult: *tim* (y-axis), *cwo* (x-axis), *Pdp1* (color), *vri* (size of datapoint). Initially expression of all four genes in ISCs (first column of graphs) is low but increases during maturation. This pattern is also seen in ECs (third column). EBs and EEs do not show these increases, cells remain clustered together at low levels of expression of all four genes. **D** PupCs also show low clock gene expression. See Supplementary Information for correlation matrices of multidimensional plots. Related to Supplementary Fig. 5, Supplementary Data 1–4. ISC intestinal stem cell, EB enteroblast, EC enterocyte, EE enteroendocrine cell, PupC pupal cell.

*Pdp1*, Fig. 4C). This analysis provides a set of potential candidates that regulate the initiation of circadian clock activity.

To test whether these and other hormone nuclear receptors regulate clock development, we performed a screen by depleting genes in a cell-specific manner. Using our scRNA-seq analysis as a guide, we depleted components of hormone signaling in ISC/EBs (*esg^TS*, 17 genes) using RNAi in *Clock^TIM* reporters during pupation (Fig. 5A and Supplementary Data 5). We reasoned that the loss of positive regulators of clock differentiation would delay Clk/cyc activity, while negative regulators would hasten it. We found that (as predicted by SCENIC) Clk/cyc activity was reduced by *Hr78*, *Hnf4*, and *gce* knockdown, however, others such as *Met*, and *ftz-f1* did not (Fig. 5A). Several other nuclear receptors were identified that also decrease Clk/cyc activity, but none were found that show an increase. To further test the requirement for hormonal signaling in activating clock gene expression, we overexpressed a subset of nuclear receptors identified

in both the SCENIC analysis (Fig. 4B) and tested by RNAi (Fig. 5A). None of these were sufficient to accelerate Clk/cyc activity in pupal stages, suggesting that these pathways are required for circadian maturation but alone are not sufficient to prematurely drive clock emergence (Fig. 5B). However, after eclosion Clk/cyc activity is significantly higher when *Hnf4* or *Hr78* are overexpressed, suggesting an important role in regulating intestinal clock development (Fig. 5B). Of note, EC reporter expression is also increased when these genes are overexpressed in ISCs, possibly due to the higher clock activity in their precursors. Since common regulatory pathways might affect subsequent clock activity in differentiating ECs as well, we retested EC-specific knockdowns of the same genes tested in ISC/EBs. Consistent with ISC/EBs, components of *Hnf4* signaling in ECs decreased Clk/cyc activity on the first day after eclosion, compared to the control (Fig. 5C).

Our SCENIC analysis also suggests that *Pdp1* may be a factor, however, *Pdp1* loss (*Pdp1^3135* mutant[74]) does not affect initiation of Clk/

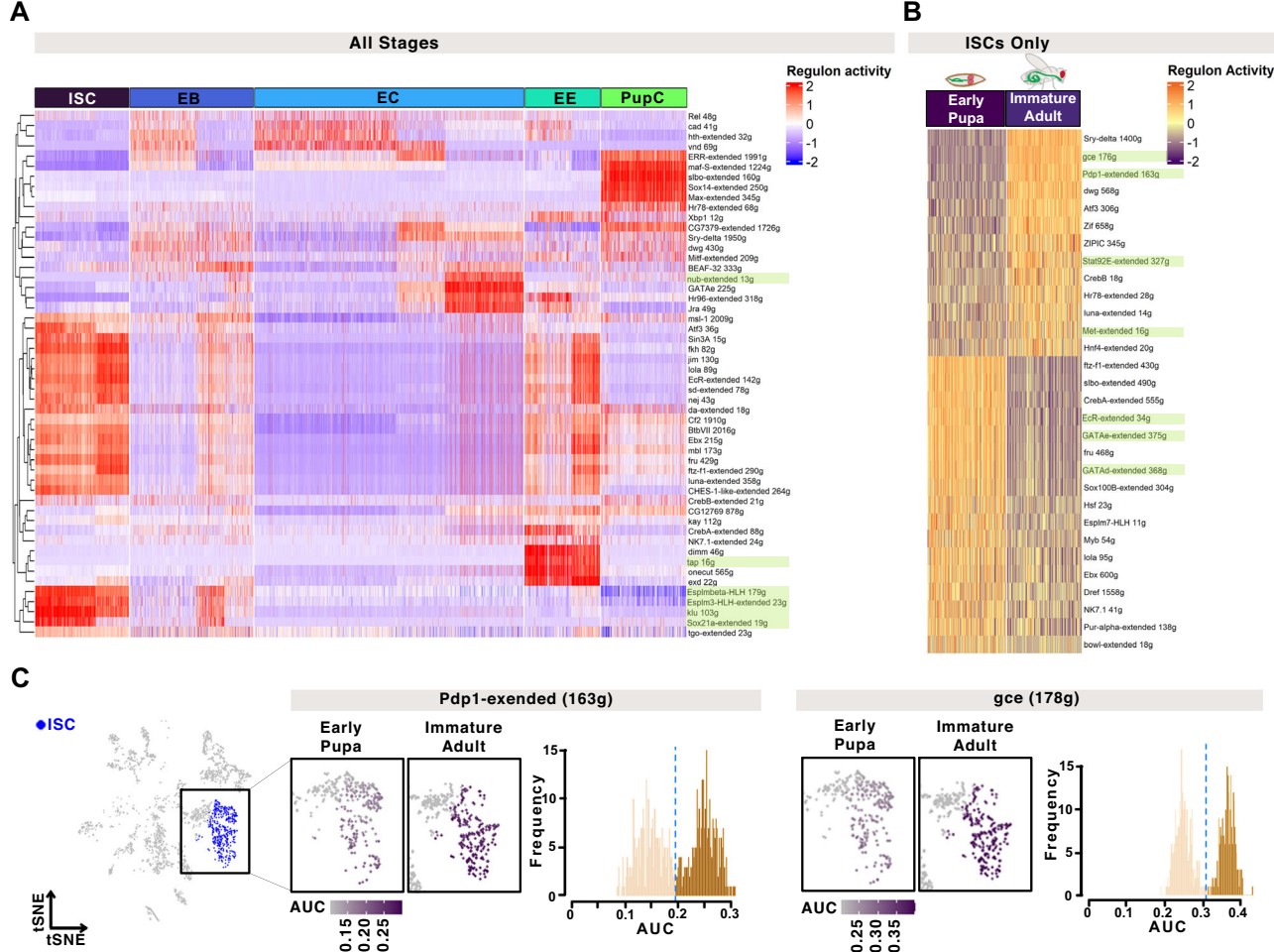

**Fig. 4 | Transcriptional analysis of intestinal clock development. A** SCENIC analysis of differentially expressed regulons that distinguish individual cell populations show clear changes in the transcriptional programs of different intestinal cell types (*klu* and *Sox21a* in ISCs, *nub* in ECs, *tap* in EEs). **B** Differential regulon expression focusing on the pupal-to-adult transition in ISCs shows that specific transcriptional pathways change during ISC development (*gce/met, Pdp1, Stat92E* are increased; *EcR, GATA* are decreased). Scaled AUC values are shown, ##g indicates the number of genes enriched in each regulon's network. **C** The tSNE plot for early pupa and immature adult cells highlighting the ISC cluster used for SCENIC analysis, AUC values shown on a tSNE and in histograms showing population distributions based on regulon expression for the nuclear receptors *gce* and the clock gene *Pdp1*. Related to Supplementary Fig. 6 and Supplementary Data 4.

cyc activity (Supplementary Fig. 6A). This suggests that *Pdp1* is not required for clock development, consistent with a role for *Pdp1* in regulating clock outputs without being necessary for *Clk* expression[75]. Together, these data are consistent with the notion that multiple hormone-signaling pathways in the intestine during pupal development regulate the emergence of the clock and highlight *Hnf4* signaling of particular importance to circadian clock development.

**The EC lineage reveals heterogeneity in clock activity during differentiation**

EBs are progeny of ISCs that differentiate into ECs[23,24], but do not express clock genes robustly (Fig. 3C). We therefore tested clock gene expression in the mature adult ISC to EC lineage. To distinguish between the steps in differentiation, we re-clustered and ordered adult cells from ISCs (cluster 0), EBs (clusters 2, and 3), and three EC populations (clusters 7, 8, 14) to reconstruct these stages in differentiation (Supplementary Fig. 6B). We then followed the longest lineage (ISC-0 to EC-7) to determine whether clock gene expression changes during EC differentiation (Fig. 6A). The expression of a control housekeeping gene, *RpL32*, is present in all clusters, whereas the gene *klu* that correlates with the ISC identity[45,76], and the EC marker gene *Amy-p*[45] are enriched in their respective populations (Fig. 6B). Clock gene expression was mapped on the same lineages and is reduced

during the different stages of EB differentiation: *Pdp1, tim*, vri, and *cwo* are initially high in ISCs, then temporarily reduced in the EB populations, re-emerging in ECs when these complete differentiation (Fig. 6B, C). Since nuclear receptors play a role in activating Clk/cyc during development, we tested the expression of hormone signaling components in ISC lineages. *Hnf4, Eip75B*, and *gce* also show high expression in ISCs, low in EBs, returning in ECs (Supplementary Fig. 6C). This suggests that nuclear receptors are correlated with clock activity during adult ISC differentiation.

These data indicate that clock genes are repressed in the ISC-to-EC lineage, hence we predicted that individual ISC clones should show Clk/cyc active and inactive cells. We used CD2-flp/out flies, where each CD2-marked clone consists of an initial marked ISC and all of its progeny, to lineage trace individual ISCs with a *Clock^TIM* reporter. The clones revealed large GFP+ ECs, and a mixture of small GFP+ and small GFP- cells, consistent with the idea that a small GFP + ISC produces small GFP- EBs that go on to differentiate into large polyploid GFP+ ECs (Fig. 6D and Supplementary Fig. 6D, E)[28]. We further tested this idea by examining Clk/cyc activity specifically in only ISCs (*esg^TS*, *Su(H)GBE-Gal80>mCherry*), only EBs (*Su(H)GBE>mCherry*), or only ECs (*Myo1A^TS>mCherry*). We predicted that EBs should show lower Clk/cyc activity than their founder cells or their fully differentiated progeny. Indeed, Clk/cyc activity is present in ISCs, absent in EBs, and highest in

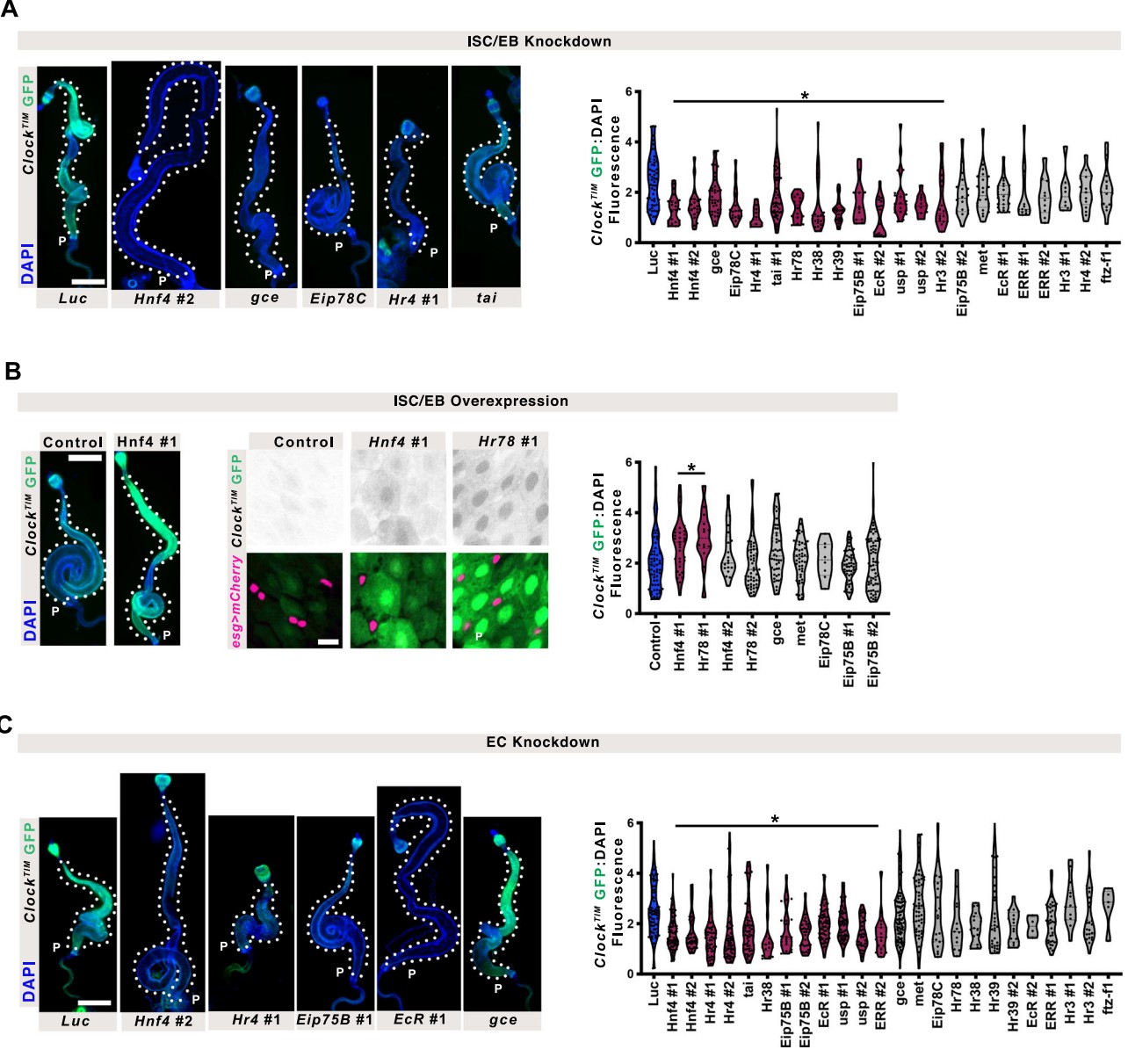

**Fig. 5 | Hormonal regulators of intestinal clock emergence. A** Knockdown of nuclear receptors in ISC/EBs show that loss of hormone signaling transduction decreases Clk/cyc activity. Scale 500 μm. DAPI marks nuclei. One-Way ANOVA *p* < 0.0001. **B** Overexpression in ISC/EBs of candidates shows that *Hnf4* and *Hr78* upregulate Clk/cyc activity compared to control midguts. Scale 500 or 10 μm for whole gut and close-up images, respectively. DAPI marks nuclei. One-Way ANOVA *p* < 0.0001. **C** Knockdown of nuclear receptors in ECs shows loss of hormone signaling transduction also affects Clk/cyc activity in ECs. Scale 500 μm. DAPI marks

nuclei. We also noticed that EC knockdown of *ftz-f1*, and ISC/EB overexpression of Eip78C, fewer flies eclosed indicating a developmental role for these genes. In all graphs: Lines show median and quartiles. One-Way ANOVA *p* < 0.0001, multiple comparison, *p*-value < 0.05 (significant groups are shown in maroon and marked with an asterisk) compared to the control (*Luc* shown in blue) are shown. Scale 500 μm. DAPI marks nuclei. Representative images of two replicates. Full statistics are shown in Supplementary Information. Related to Supplementary Fig. 6 and Supplementary Data 4–5. Source data are provided as a Source Data file.

ECs (Fig. 6E–G). These data are consistent with our scRNA-seq analysis and suggest the transient disappearance of circadian clock activity during lineage commitment of ISC to ECs in the adult intestine (Fig. 6H).

**Photoperiod and feeding synchronize the maturing intestinal circadian clock**

Our results indicate that the intestinal circadian clock gene expression is established during development, however, it is not clear what factors cause the intestine to exhibit synchronous circadian rhythms. The intestine completes its growth and development after adult emergence from pupation[35,48], while it is being synchronized to the environment (Fig. 2B, C). We have previously shown that the *Drosophila*

intestinal clock is synchronized by photoperiod light cycles, with food ingestion also playing a modulating role in timing daily rhythms[28]. We first asked if photoperiod would be sufficient to synchronize the immature intestinal clock. Flies were completely starved for the first three days of adulthood following pupation and provided food only at ZT0 on adult day 4 when clock synchronization is complete. Even though intestinal size was reduced by lack of nutrients (Supplementary Fig. 7A, B), Clk/cyc activity was robust, and synchronized rhythms emerged one day earlier (day 3 instead of day 4, compare Fig. 7A to Fig. 2C), suggesting that feeding and/or intestine growth delays clock entrainment.

We then performed the opposite experiment, in which photoperiod is absent and feeding is present, by using *cry^{01}* mutants that are

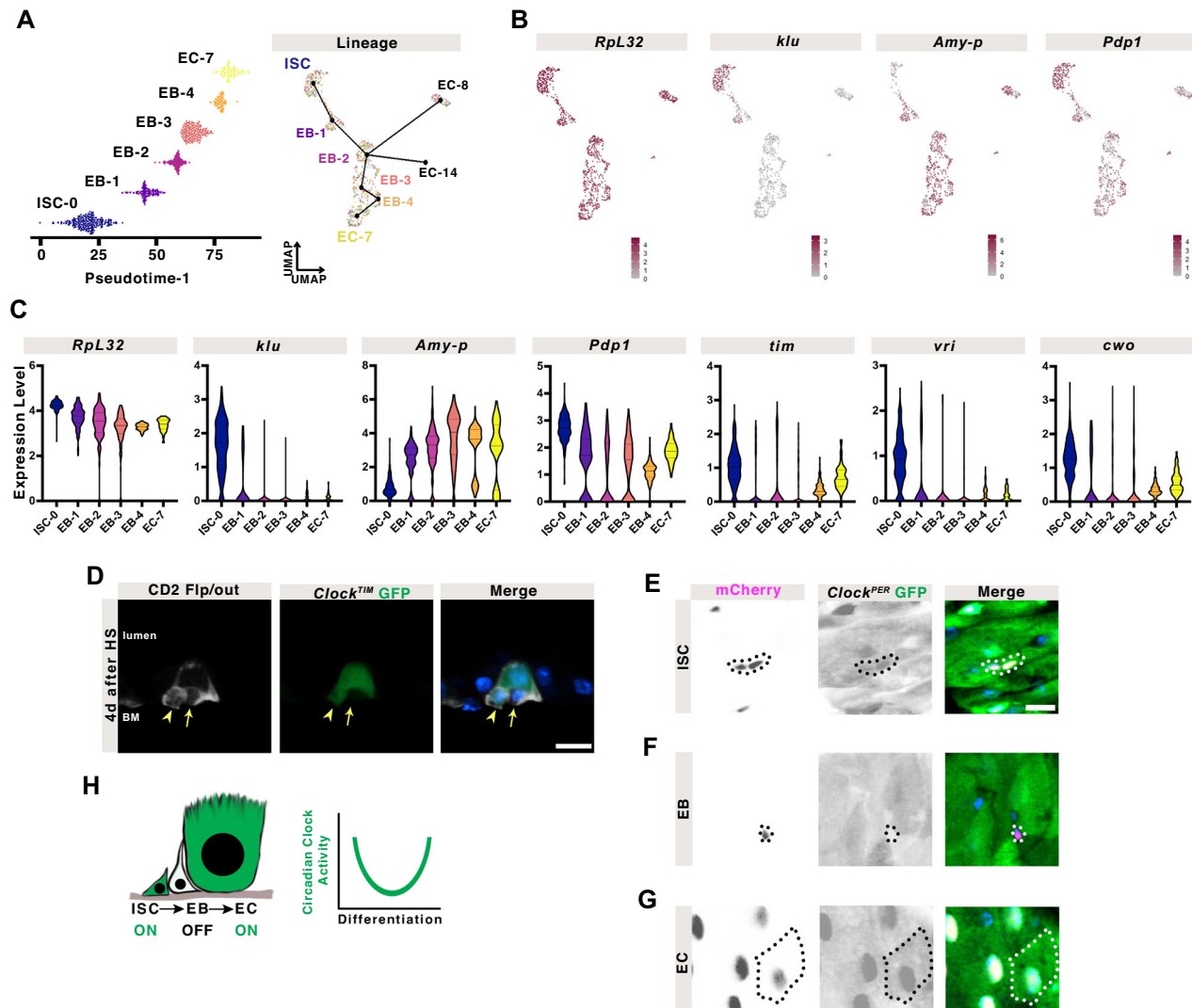

**Fig. 6 | The EC lineage reveals heterogeneity in clock activity during differentiation. A** A subset of clusters from the mature adult intestine (0,2,3,7,8,14) were assembled into a lineage from earliest cells (ISC = cluster 0) to differentiated ECs in pseudotime. See Supplementary Fig. 6B for more details. **B** Expression of a housekeeping gene, *RpL32*, shows it is expressed in all cell populations, whereas ISC-specific *klu* and EC-specific *Amy-p* are restricted to their respective populations. Mapping clock genes such as *Pdp1* shows their expression in these lineage changes. **C** Graphs show cellular expression levels in this lineage of *RpL32*, compared to cell-specific genes (*klu* and *Amy-p*), and clock genes (*Pdp1*, *tim*, *vri*, *cwo*). *RpL32* is expressed throughout, *klu* only in ISCs, *Amy-p* only in differentiating EBs and ECs. Clock genes show initially high expression in mature ISCs which lowers in the transient differentiating EBs, to increase again in the EC. Lines indicate median and

quartiles. Kruskal-Wallis test, full statistics are shown in Supplementary Information. **D** Flp/out clones show a mixture of basally located GFP+ and GFP- small cells, suggesting that Clk/cyc activity is heterogeneous in either ISCs and/or their progeny. Arrowhead marks GFP+ cell, arrow marks GFP- cell. Scale bar 10 μm. DAPI marks nuclei. Additional clones are shown in Supplementary Fig. 6D, E. Using mCherry expression to mark (**E**) ISCs, (**F**) EBs, and (**G**) ECs, specifically, Clk/cyc activity (*Clock^{PER}*) is present in ISCs, absent in EBs, and strongest in ECs. Outline indicates cell of interest marked with mCherry. Scale bar 10 μm. DAPI marks nuclei. An example of each mCherry+ cell is outlined. **H** Schematic summarizing the observed decrease in Clk/cyc activity and clock gene expression during differentiation. Representative images of two replicates. Related to Supplementary Fig. 6 and Supplementary Data 4.

unable to detect light to synchronize clock function[77]. We predicted that feeding would synchronize circadian rhythmicity. Without photoperiod, *cry^{01}* intestines exhibit arrhythmic Clk/cyc activity during the first three days but show Clk/cyc activity at day 4 onward similar to wildtype controls (Fig. 7B compare to Fig. 2C). Since light-responsive feeding behavior occurs in *cry^{01}* flies[78] these data suggest that feeding alone can also complete circadian clock synchronization in the intestine. To further test this, we restricted feeding in *cry^{01}* flies post-pupation to the morning only (ZT0-6), evening only (ZT5-11), or subjected them to alternate days of feeding and starvation to disrupt feeding rhythmicity. We predicted that the timing of food consumption would alter the maxima and minima of intestinal circadian

rhythms. Indeed, under these conditions the rhythms of *cry^{01}* intestines were shifted to peak around 6 h after feeding, or rendered non-circadian when disrupted (no clear 24-h period) (Fig. 7C). In contrast, control flies that have the ability to detect photoperiod do not show a shift in Clk/cyc activity on restricted feeding, suggesting that photoperiod is dominant over feeding if both environmental cues are detected (Supplementary Fig. 7C). We further tested if periods of fasting would affect clock rhythms once these are established in adults and noted that these do perturb the clock if photoperiod is detected (Supplementary Fig. 7D, E). We conclude that the combination of daily photoperiod and feeding cycles together synchronize the timing of intestinal circadian transcription cycles during clock maturation.

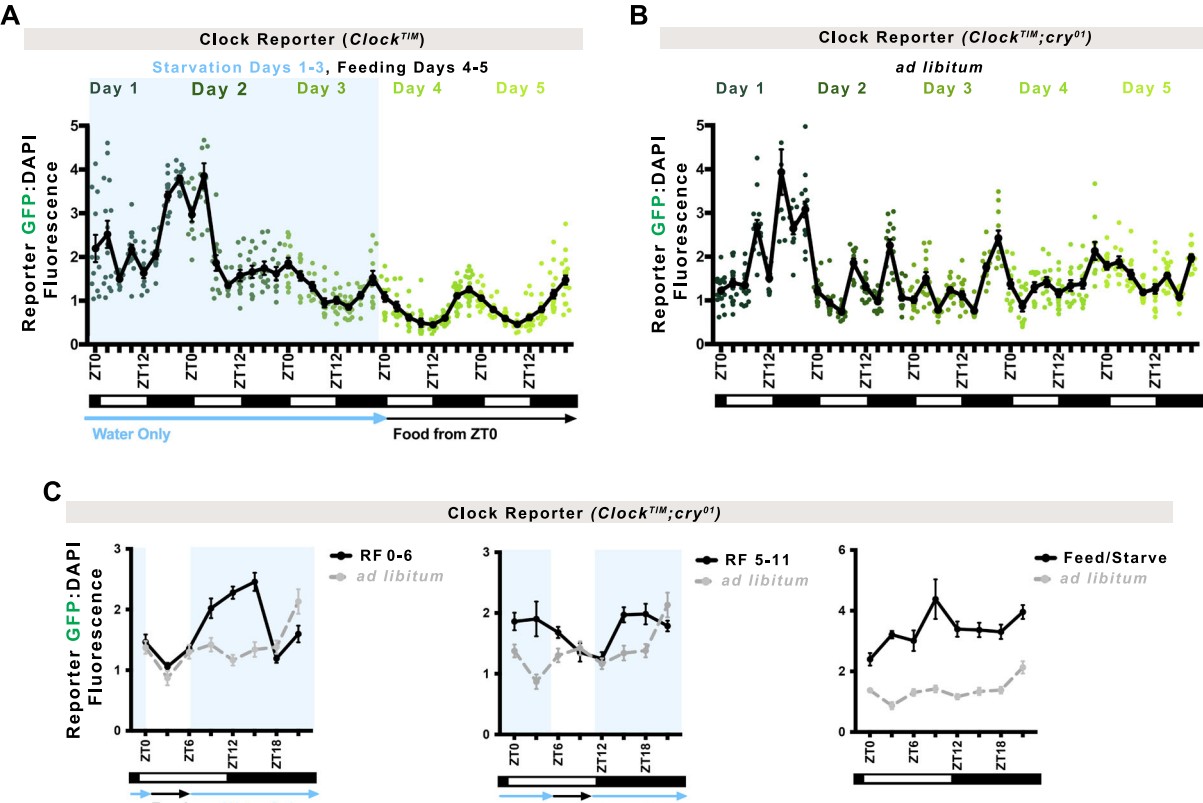

**Fig. 7 | Photoperiod and feeding synchronize the maturing intestinal circadian clock.** **A** *Clock^TM* flies that receive only water for the first three days after pupation show robust rhythmic Clk/cyc activity earlier: by day 3 rather than day 4, suggesting the absence of food more quickly synchronizes a rhythmic clock. Line shows mean. Error bars indicate ±SEM. Each point represents one intestine, $n = 595$ intestines over two independent replicates. One-Way ANOVA $p < 0.0001$.
**B** *Clock^TM;cry^01* mutant flies that cannot sense photoperiod light display arrhythmic Clk/cyc transcriptional activity until day 4 when these are established, similar to controls. This suggests food in the absence of photoperiod can also synchronize a rhythmic clock. Line shows mean. Error bars indicate ±SEM. Each point represents one intestine, $n = 570$ intestines over two independent replicates. One-Way ANOVA $p < 0.0001$. **C** Alternating days of feeding and starving (water only) shows arrhythmic Clk/cyc activity whereas restricted feeding for 6 h a day in *Clock^TM;cry^01* mutants synchronizes intestinal Clk/cyc rhythms that peak ~6 h after feeding. *Ad libitum* measurements (control) from Fig. 7B are replotted in gray to show difference. Line shows mean. Error bars indicate ±SEM. Two-Way ANOVA Feed/Starve $p = 0.0047$; RF5-11 and RF0-6 $p < 0.0001$. Fasting/Starvation times outlined in light blue rectangle(s). Full statistics are shown in Supplementary Information. Related to Supplementary Fig. 7. Source data are provided as a Source Data file.

## Discussion

There are many questions about how circadian rhythms develop in the body[10,79] and, in particular, connections between circadian rhythms and stem cells are poorly understood[80,81]. We show that as the intestine grows and differentiates clock gene transcription is negatively regulated (Figs. 1, 3); ISCs differentiating into EB precursors also do not express clock genes (Fig. 6). Our study sheds light on two processes: 1) the emergence of circadian clock function in the stem cells of the intestine and 2) suppression in their immediate progeny as these establish a differentiated cell state. These data suggest that the clock and differentiation are incompatible in the intestine and are reminiscent of a recent report showing clock function is incompatible with mouse somite development[82]. Our study characterizes the development of the circadian clock in the intestine but this framework may be extended to other organs of the body.

Circadian clock development has been tested in several animal species. In zebrafish, clocks emerge during the first stages of embryogenesis[5], similarly, in *Drosophila* clock emergence was shown to occur early in development[16]. However, these initial studies of *Drosophila* clock development focused on circadian rhythms in the brain rather than other tissues in the body[12,16]. It was recently found that *Drosophila* larval tissues do not express the clock gene *cry* until adulthood[39], and circadian control of the prothoracic gland also occurs in pupation[83], consistent with our results (Fig. 1). Of note, we find that

Clk/cyc activity is initiated in the spiracles and prothoracic gland during larval development, and the heart during pupation (unpublished observations KP, PK). This suggests that circadian timing in the *Drosophila* central clock emerges earlier and timing in other tissues emerges later. In chicks, even though clock expression occurs early in development, it is patchy during early stages before becoming more widely expressed[84]. In mice and rats, where circadian development is best studied, early embryonic cells do not have clock rhythms[85,86]. The central pacemaker in the brain emerges during late embryogenesis[9,87,88], and other tissues such as the heart[89], kidney[90], and colon[7] emerge even later. These mammalian tissue clocks are present just before birth and their synchronous rhythms mature at different rates postnatally[91], particularly in the colon that (like our study) shows the intestine has a relatively slow clock to develop in the body[7]. Overall, data from a wide variety of species are consistent with our observations. Circadian clock function is also tissue-specific, and thus its emergence may be tissue-specific, eventually coalescing into complete circadian rhythmicity throughout the body at the appropriate stage of development. The larval stages where insects eat almost continuously may be incompatible with a daily timing system in the digestive tract that dictates activity/rest cycles.

Our study is the first to examine the emergence of animal tissue clocks at the single cell level, revealing that stem cells are the first to pass clock status to their differentiating progeny. The transient loss of

clock gene expression as cells transition between discrete fates has been previously overlooked when using population-based methods such as RNA-seq or RT-qPCR. Thus, our study provides a paradigm of how clocks may develop in non-neural tissue cells after the central clock is established. We have used the expression of several clock components simultaneously to estimate the developmental timing of clock rhythms. It is important to recognize that the expression of clock genes alone does not confirm such rhythms are present. Indeed, we find that clock gene expression is present in the early adult intestine but is not robustly synchronized and rhythmic until several days later (Figs. 1–2). The early clock in the zebrafish circadian system also takes 3–4 days to be established, with the secondary feedback system and output genes rhythmic later in development[92,93]. This 2–4 day delay between gene expression and daily cycling also appears in the mouse kidney[90] and central clock in the brain[9].

Elegant studies in mouse embryonic cells have shown that translational regulation of *Clock* (the mouse ortholog of *Clk*) is a rate-limiting step in circadian development[89]. Our results are in agreement with this (Fig. 1H), however, in *Drosophila* we identify the component *Pdp1* as another potential factor (Figs. 1H and 4C). This is consistent with a Clk-based transcriptional model because Pdp1 positively regulates *Clk* expression, hence *Clk* expression and transcriptional activity may be a conserved circadian component that is required for rhythmicity. For instance, since Pdp1 and vri maintain rhythmic *Clk* expression[20], we note that their developmental maturation over days 1–4 may explain the initially low amplitude rhythms in Clk (Fig. 2A). Although we find that the loss of *Pdp1* does not affect the timing of intestinal clock emergence, in *Drosophila*, it has been shown that ectopic expression of *Clk* can prematurely induce clocks outside the pacemaker[94,95]. Future work to characterize this potential mechanism in ISCs will be informative, as it may be a switch that initiates daily tissue timing in developing tissues at different times. Previous studies have shown that in the mouse liver, *Hnf4* is essential for Clock/Bmal1 rhythmic activity[96], and our results suggest that it also regulates the development of the *Drosophila* intestinal clock (Fig. 5). In addition, environmental factors such as the microbiome might play a role in maturing both the tissue and its circadian rhythms, although we note that it was recently reported that in the adult midgut the microbiome does not appear to show rhythmicity[97].

This dataset provides single-cell characterization of insect intestinal metamorphosis and circadian clock development. Because insects utilize circadian rhythms to coordinate their pupation timing[98], these results are informative in understanding the fundamental biology of these animals. Our analysis with SCENIC[52] identified key transcription factors involved in the development of this organ, showing the existence of transitory Pupal Cells (PupCs) that express markers of early differentiation such as *Sox14*[53]. Using scRNA-seq, we were able to recover the main cell types in the intestine at all three stages and found markers to identify ISCs, EBs, EEs, ECs, and visceral muscle (VM) that are consistent with previous *Drosophila* intestine sequencing studies[35,44–47] (Supplementary Data 4). We were not able to identify a population of enteroendocrine mother cells although the earliest ISC population includes cells enriched for *piezo*, a marker of the EE lineage[36], therefore it is likely that these cells are located in the ISC cluster (Cluster 0, Fig. 3A). We have annotated a population of pupal cells, PupCs, that are unique to the pupal intestine and display similarities to differentiated cells including expression of *MtnB* and *MtnD* characteristic of the central intestine, Supplementary Data 4) suggesting that these cells contribute to intestine development consistent with the transient pupal midgut described previously[25].

By testing clock development at the single cell level, we found that the circadian clock program is downregulated during the dynamic cellular changes that occur during differentiation. These results are counterintuitive given that the clock is thought to emerge during embryogenesis, and has in fact been demonstrated to arise in differentiating embryonic stem cells[9,88], and organoids[99] in vitro. Indeed, several studies have shown circadian clocks regulate cellular differentiation itself[100–102]. We therefore propose that during the transient states of cellular fate specification, clock gene expression is disrupted. A parsimonious explanation for this is that the transcriptional and epigenetic reorganization that occurs during differentiation[103–105] simply interferes with rhythmic circadian transcription. Interestingly, a recent study observed a similar pattern of circadian disruption in thymocyte differentiation where a Per1-Venus reporter exhibits reduction in cellular stages between thymocyte precursor and fully differentiated cell[106]. In line with our data, this suggests that clock function requires a certain level of epigenetic stability that is not compatible with the rapid cellular fate changes that occur during development or during stem cell-driven regeneration. The *Drosophila* midgut provides an opportunity for future investigation of these interactions and highlight a state where cells in the body are resistant to circadian rhythms. Because both ISC differentiation (this study) and somite differentiation[82] involve cell fate commitment through Notch signaling, it will be of interest to see if these principles extend to other developmental pathways and/or confirm the precise mechanisms linking the clock and the Notch pathway in the future.

In human beings, circadian rhythmicity arises postnatally: most infants take several months to establish coherent daily timing[6]. This is likely to reflect the complexity of establishing robust inter-cellular and inter-organ synchrony that involve coordinating environmental signals with tissue-specific clocks. Our study reveals how circadian rhythms are established in complex developing tissues and provides a system in which to investigate the birth of the circadian tissue pacemaker.

## Methods
### Fly Strains
*Drosophila melanogaster* strains in this study include: *Clock^PER^*, *Clock^TIM^* described in[28]. From Jadwiga Giebultowicz: *CantonS* and *per^01^*(isogenic with *CantonS*). From Chrysoula Pitsouli *esg-Gal4 (II)*, *tubGal80^TS^(III)*. From Yong Zhang *PER-AID-GFP*. From Steve Jean *Su(H)GBE-Gal80, tubGal80ts(III)*. From Amita Sehgal *Pdp1^3135^(III)*. From Bloomington: #38424 *UAS-mCherry (III)*, *UAS-mCherry(II)*, *myo1A-Gal4 (II)*, #42563 *Valium20-cyc(II)*, *cry^01^*, #76317 *CRY-GFP*, #4412 *act > y + >Gal4,UAS-CD2*, RNAi lines: #50712 and #58286 *EcR* (#1 and #2), #43231 *Eip75B*, #26718 *Eip78C*, #61937 and #50868 *ERR*, #27659 *ftz-f1*, #61852 *gce*, #29375 and #64988 *Hnf4*, #51442 and #27254 *Hr3* (#1 and #2), #29376 and #29377 *Hr38*, #33624 and #27086 *Hr39*, #54803 and #31868 *Hr4*, #31990 *Hr78*, #31603 *Luciferase*, #26205 *met*, #36095 *tai*, #36729 and #27258 *Usp* (#1 and #2). From FlyORF: F001915 *gce*, F000656 *Eip78C*, F000144 and F003392 *Hnf4* (#1 and #2), F000423 and F003303 *Hr78* (#1 and #2), F004847 and F002207 *Eip75B* (#1 and #2), F003474 *met*. Flies were balanced with *yw;IF/CyO;MKRS/TM6B*. Experimental genotypes are provided in Supplementary Table 1. Flies were raised on cornmeal-glucose media (1.2% w/v yeast, 0.7% w/v soyflour, 5% w/v cornmeal, 0.4% w/v malt, 0.4% v/v agar, 5.3% v/v glucose with propionic acid and tegosept) at 25 °C with 12:12 light/dark photoperiod (lights-on at ZT0). For free-running experiments flies were shifted to constant darkness (DD) during the dark-phase of the final day of LD, CT0 (Circadian Time 0) is the time when lights are expected to turn on. For larval to adult analysis, flies were only allowed to lay eggs for 5 h in the morning (ZT0-5). For adult experiments, only flies that eclosed overnight (ZT12-0) were included, no difference was noted between ZT11-12, ZT23-0, and ZT0-1 eclosion groups. Heatshocks for Flp/out clones were done with adult flies 5–14 days old for 5–10 min at 37 °C. For RNAi and overexpression screening (Fig. 5), flies were raised at 18 °C until pupal stages then vials were shifted to 29 °C and both males and females were collected. Age ranges from first larval instar to adults (day 1–14).

## Food restriction

Pupa were transferred to vials containing a piece of water-moistened filter paper using a paintbrush. Flies eclosed onto this filter paper and were kept on water after eclosion either (1) until adult day 4 at ZT0 when flies were transferred to vials containing normal food or (2) for 6 h (from either ZT5-11 or ZT0-5) each day after eclosion. The alternate days of feeding and starvation (Fig. 7C) were done by allowing flies to eclose onto water-moistened filter paper, similar to the restricted feeding, then at ZT0 were flipped onto food ("F"), then the next morning at ZT0 back onto filter paper with water for starvation ("S"), then repeated F (day 3), S (day 4). This alternating cycle was necessary, since unlike the controls the $cry^{01}$ flies did not survive the starvation for 2–3 days.

## Developmental staging

Larva were roughly staged using the following times: 1 day AED (after egg deposition) for L1, 2 days AED is L2, 3–5 days AED is L3 (the day AED may be noted e.g., L3 day 3 refers to three days AED). Early stage pupa were analyzed at the red-eye stage (day 8 AED), and late stage pupae were analyzed at the gray to black wing stage (day 9 or 10 AED)[107]. These stages were verified visually to account for individual variation in developmental timings. The first day after eclosion is referred to as adult day 1 (day 9 or 10 AED).

## Dissection

Flies were euthanized in 70% EtOH and then intestines were dissected using fine forceps in 1xPBS. For RNA extractions and single cell analysis, the hindgut (anterior to the Malpighian tubules) and the foregut were removed during dissection to ensure only the intestine was sampled. Dissections were completed in <1 h. Analysis from larva to adult included both males and females but unless otherwise noted, only female flies were analyzed.

## Tissue preparation/antibody staining

Intestines were fixed in 4% paraformaldehyde for 40 min, washed twice in 1xPBS and stained with DAPI (1:5000 in 0.2% Triton-X100 (in 1xPBS)) for 5 min. The intestines were washed twice in 0.2% PBS-T and then mounted on slides using AlexaProLong Gold Antifade Reagent (Invitrogen). For antibody staining, intestines were fixed for 2 h in 4% paraformaldehyde and then washed twice in 1xPBS and blocked for 20 min in 1% bovine serum albumin in 0.2% PBS-T (BSA). The intestines were then incubated in primary antibody (PER (1:1500), histone (1:2000), GFP (1:2000), rat-CD2 (1:2000)) in BSA for 2 h, washed twice in 0.2% PBS-T, incubated in secondary antibody (1:2000) with DAPI (1:5000) in BSA for 1 h and then washed twice in 0.2% PBS-T (at 4 °C) and mounted on slides. Cry-GFP was stained 1-h prior to lights on (ZT23) to avoid light-induced degradation of $cry$[39].

## RT-qPCR

Approximately 15 female flies of *CantonS* were dissected in 1XPBS before being transferred to tubes containing RNAlater (Qiagen) on ice. RNA was extracted using the RNAMini Kit (Qiagen) with RLT buffer (Qiagen) using a Bullet Blender (Next Advance) for homogenization. cDNA was made using ISCRIPT RT Supermix (Bio-Rad) and RT-qPCR was performed using SYBR green (Bio-Rad) using the Viia7 PCR plate reader, primers: per-F TCATCCAGAACGGTTGCTACG, per-R CCTGAA AGACGCGATGGTGT[27], tim-F CCAGCATTCATTCCAAGCAG, tim-R GCGTGGCAAACTGTGTTATG[27], Gapdh-F CCAATGTCTCCGTTGTGGA, Gadph-R TCGGTGTAGCCCAGGATT, Clk-F CAAGTTTGGCCTCTGGC TCTC, Clk-R TACAACTAGCTCTGGGCTTCCG, Pdp1-F GAACCCAAGTG TAAAGACAATGCG, Pdp1-R CTGGAAATACTGCGACAATGTGG[108], vri-F TGTTTTTTGCCGCTTCGGTCA, vri-R TTACGACACCAAACGATC GA[108], cyc-F GCGCTGATGGAGTCTCACAAG, cyc-R GTAGCTGTTGTCC TTGCACCG, cry-F CACCGCTGACCTACCAAA, cry-R GGTGGAAGCCC AATAATTTGC[109], dco-F TTGGGAGGAGGGTTAGCAG, dco-R TTACAAT GTGGGTGCCTTGC.

## scRNA-seq

Flies from *CantonS* were euthanized in 70% ethanol in DEPC-treated water at ZT0. Every 10 min during the 1 h dissection time, the intestines were transferred from the 1XPBS well to ice-cold 1% BSA. Once dissections were complete, the intestines were transferred to a drop of 1XPBS on the back of a dissection plate and cut into small pieces using microscissors. The fragments were then transferred to a 1.5 mL tube to a total volume of 350ul (200uL for pupa). Elastase was added to a concentration of 1 mg/mL and incubated at 37 °C for 40 min pipetting up and down 30–40 times every 10 min to dissociate the cells, then 10% BSA was added to a final concentration of 1% BSA. The samples were centrifuged at 300 x g for 15 min at 4 °C, then the pellet was resuspended in 200ul of 0.04% BSA and filtered with a 70um filter centrifuging at 300 x g at 4 °C for 1 min. Orbitrap Density Gradient was used to sort the cells and then the interface was washed with 0.04% BSA. The cell density was determined with a haemocytometer and diluted with 0.04% BSA to a density of 800cells/uL. All samples had a viability >95%. The entire preparation was completed within 2 h. The fresh samples were submitted to the High Content Analysis Core at the University of Alberta by ZT3 for 10X Genomics 3' Library preparation and to Novogene for sequencing (NovaSeq) and then aligned to the Drosophila reference genome 6.25 using Cell Ranger (10X Genomics, version 3.0.2).

## Imaging and data analysis

Slides were imaged with either a slide scanner (Zeiss, Axio Scan Z.1) or confocal microscope (Olympus, IX81 FV1000 with 60x Water Immersion Lens or Zeiss Confocal with 20x lens). Whole mount imaging and quantification were done using a fluorescence microscope (Zeiss, VertA.1) or fluorescence stereoscope (Leica M205). Images were analyzed with Zen Blue Software (Zeiss, version 2.3) and processed using Photoshop (Adobe) with adjustments consistent within a panel. Fluorescence intensity measurements are ratios of GFP:DAPI taken for the whole midgut, unless otherwise specified, as previously described[110]. RT-qPCR data is normalized to *Gapdh1*. Prism (GraphPad) was used for graphing and statistical analysis. Cosinor analysis was performed using circacompare[111] and graphed using Prism (GraphPad). Schematics were prepared using Photoshop (Adobe) and Illustrator (Adobe).

Single sequencing data was analysed in R using Seurat (version 4.3.0.1)[112] and pseudotime analysis was performed using Slingshot (version 2.8.0)[113]. The original dataset of 6835 adult, 2625 pharate, 3816 pupal cells was filtered (genes must be detected in > = 3 cells, %mitochondrial <25, 200<nFeatures<4000, 1000<nCount<100000) and the remaining 1980 adult, 1197 pharate, 2013 pupal cells were used for further analysis. The datasets were integrated (reduction = rpca) and clustered (PCs = 14, resolution = 0.5) and then annotated using differentially expressed genes (method = roc). For lineage tracing 1111 adult cells consisting of clusters 0,2,3,7,8,14 were re-clustered (PCs = 9, resolution = 0.3) or 1933 pupal cells from clusters (0–6,8–14,16,17,18,21) were re-clustered (PCs = 16, resolution = 0.3) prior to lineage tracing to further subdivide these cells. Differential gene analysis was performed using roc. SCENIC analysis (version 1.3.1)[52] was performed with version 10 *Drosophila* motif set with minimum number of genes in each regulon = 10, then handed back to Seurat for differential gene expression analysis. In some cases, Prism (GraphPad, version 9) was used for graphing and statistical analysis. Images were processed using Photoshop (Adobe, version 23.4.2) and schematics were made using Photoshop and Illustrator (Adobe, version CS5).

Statistical tests were performed using GraphPad Prism and can be found in Supplementary Information.

**Reporting summary**

Further information on research design is available in the Nature Portfolio Reporting Summary linked to this article.

## Data availability

The scRNA-seq data generated in this study have been deposited in the NCBI GEO database under accession code GSE230572 [https://www.ncbi.nlm.nih.gov/geo/query/acc.cgi?acc=gse230572]. The alignments were made using Drosophila reference genome 6.25 and are publicly available as of the date of publication. Source data are provided with this paper.

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

## Acknowledgements

We are grateful to the members of Karpowicz, Foley, and DeVeale labs and the technical staff at the Universities of Windsor and Alberta for their insightful advice and expertize. K.P., A.Z., and P.K were funded by the Canada Foundation for Innovation, the Ontario Research Fund, and the Natural Sciences and Engineering Research Council of Canada (RGPIN-2020-04252 P.K.). B.D. was funded by the Natural Sciences and Engineering Research Council of Canada (RGPIN-2023-04028 BD). M.S., R.W., and E.F. were funded by Canadian Institutes of Health Research (MOP77746 EF). To the authors whose research we could not cite due to space limitations, we offer our apologies and thanks.

## Author contributions

Investigation: K.P., A.Z., M.S., and R.W.; Formal analysis: K.P., A.Z., R.W., and B.D.; Writing – Original Draft: K.P. and P.K.; Writing – Review & Editing: K.P., B.D., E.F., and P.K.; Conceptualization: P.K.; Supervision: E.F. and P.K.; Funding acquisition: E.F. and P.K.

## Competing interests

The authors declare no competing interests.
