## [Peer Review File · Nature Communications]

The Emergence of Circadian Timekeeping in the IntestineREVIEWER COMMENTS

Reviewer #1 (Remarks to the Author):

In this manuscript Parasram and colleagues examine the timecourse of appearance of clock function in the *Drosophila* intestine. Using a variety of different approaches they show that clock genes are not expressed in the intestine until the emergence of the adult fly, and that a robust circadian rhythm of clock gene expression does not appear until day 4 of adult life. Furthermore, they show that a mature clock first appears in the intestinal stem cells. Interestingly, they find that there is a transient loss of clock gene expression as cells transition between discrete fates. Finally, they show that the combination of daily photoperiod and feeding cycles together synchronize the timing of intestinal circadian transcription cycles during clock maturation.

The work reported here provides a careful description of the timecourse of gene expression in different cell types as the intestine develops from the larval to the adult stage. And, because the key clock genes are not expressed until adult emergence, it is clear from their analyses that there is no functional clock in the intestine until adulthood. Beyond this, however, this reviewer is not convinced about the exact timecourse of emergence of clock function in this tissue.

Specifically:

1- The results shown in Fig. 2A do not preclude the possibility that there is a functional clock by day 1, but that the clocks of different cells are not synchronized, leading to a lower overall rhythmicity. To address this question the authors would need to follow individual cells, and determine when a daily rhythm emerges. For this the tissue could be cultured and the intensity of their clock reporters in individual cells followed over time. (If I understood the legend correctly, in the timecourses shown in Figs. 1B and 1C each intestine contributed a single measurement, which I assume is a measure of average fluorescence intensity over an area of the intestine)

2- The statement (legend to Fig. 2) that “Clk/cyc transcriptional activity is initially stochastic and shows mature rhythms only by day 4 for ClockPER with a peak around ZT0 and trough around ZT12.” is not convincingly supported by the data. By eye, it looks like Fig. 1C shows a daily rhythm starting on day 1 with a daily minima at around ZT12. Regardless of what is seen by eye, it is really not possible to assert that rhythmicity does not emerge until day 4 absent a quantitative frequency analysis. Such a statement would also require having at least an extra day’s worth of data, so as to have 2 days of robust rhythmicity (days 4-5) to contrast with the situation on days 1-2, for instance. I was also confused by the statistics provided (in Supplemental information), which for Figure 2B and 2C indicate the values of an ANOVA analysis. An ANOVA analysis only tells us that there are statistically significant differences among the values for different times, but this does not inform us as to whether there are daily rhythmicities and when such rhythmicity becomes significant.

3- Finally, the emergence of circadian function would also require records to be obtained under dark:dark (DD) conditions. The only such record is shown in Fig. S3, for which, incidentally, it isn’t clear when the measurements were taken relative to the start of DD (ideally, measurements should be taken starting 24-48h after the beginning of DD, in case there is some residual rhythmicity). (As an aside, the legend to this figure states that “...full statistics are shown in Supplemental Information.”, yet this figure is part of Supplemental information and no statistics are shown in the “Statistics” table.)

4- Many of the data have to do with changes associated with the maturation of the intestine rather than specifically that of circadian clock function (e.g., Fig. 4 and parts of Fig. 5). In addition, and as detailed above, the “circadian” part of the results is not very conclusive. For these reasons I would suggest changing the emphasis of the study and focusing on the maturation of the *Drosophila* intestine in general rather than more narrowly on the appearance of circadian clock function in this tissue.

5- The methods section needs additional details, in particular with respect to the measurements of fluorescence intensity.

Minor comments:

- “phosphatase” should be deleted from line 106 “... the phosphatase *dbt*, a kinase that regulates...”.
- The differences in color are not evident in the symbols shown in Figure 3 (especially purple vs. blue). Please make more pronounced.

Reviewer #2 (Remarks to the Author):

In the manuscript titled “The Emergence of Circadian Timekeeping in the Intestine”, Parasram and colleagues characterized in detail the emergence and development of the circadian clock in the intestine of *Drosophila melanogaster*. In this work, researchers explored how different circadian genes are expressed during the developmental stages and in which intestinal cells these are more relevant. Using specific clock reporters, researchers showed changes in the activity and rhythmicity of TIM and PER in ISCs and their progeny cells, EB and EC. Importantly they concluded circadian rhythmicity and cell differentiation are exclusive processes, and the clock needs to be repressed in early differentiated cells (EB) to be restored in fully differentiated EC. Moreover, they demonstrated that expression of several genes started early in development (early pupa), but rhythmicity is only established in the adult after 4 days, highlighting the complexity of the circadian regulation in the gut. Interestingly feeding and photoperiod have to some extent compensatory effects, upon the lack of one or another, in influencing the establishment of circadian rhythmicity in the gut.

In summary, the manuscript demonstrates a novel approach to understanding circadian regulation during organ and tissue development and these findings contribute significantly to the existing body of knowledge in the field. This manuscript addresses a very important biological question, it is well-written and organized, making it easy to follow, ideas are clear, and figures and interpretations are well presented. Overall, data presented supports the main conclusions.

Major comments

- Clock reporter activity in ISCs is not clear from the images presented. For instance, in Figure S2F, while nuclear reporter activity is clearly observed in the ECs, it is not present in ISC/EB and therefore the claim of the authors regarding this is not supported by this evidence.
- There is a lack of information on the changes in the localization of PER at different time points. Authors should include a time course of cell specific staining of PER and/or TIM in the gut to substantiate the reporter information across intestinal development and cell types. For example, include a time course showing changes in localization of PER and/or TIM antibody comparing larvae, pupa, immature adult and

mature adult gut 4 looking at the different cell types in the gut.

- The weakest point in this manuscript refers to the functional data on molecular regulators of clock gene expression based on the scRNAseq data (Figure 4). Is any of the data presented in Figure 4D statistically significant? What is the rationale behind knocking down these various genes in ECs and ISCs/EB, when the focus, as per the authors' statement and the transcriptional information presented in Figure 4B refers to the latter cell type? Furthermore, what is the evidence leading to Bursicon signaling (rk) as a possible player here? No previous evidence has been presented on expression or functional role of rk in ISCs or ECs and therefore the rationale for these experiments is unclear. The data in this part of the manuscript needs to be re-evaluated and improved/strengthened. I suggest choosing a couple of the strongest candidates and analyse their function in reporter expression over time in ISCs, coupled with immunostaining with core clock components (e.g. PER, TIM)

- Is the microbiome playing a role in establishing the rhythmicity of the intestine during development and in the adult stages analysed?

- Is there any role of the central clock in the establishment of intestinal clock during development?

- Are all clocks in ISC synchronized or do you see different phases among them? This information could be obtained from the analysis of the ISC molecular signature in the scRNAseq. For example, if there were sub clusters of ISCs that express different levels or components of circadian genes.

- Consistent with the data presented here, Figure S5D, pdp1, is not essential for central clock expression and oscillations but for regulating circadian outputs, such as locomotion (Benito et al., 2007). Have the authors identified any function of pdp1 in gut development or rhythmic behaviours (e.g. regeneration)? The mentioned prior evidence and associated publication should also be acknowledged by the authors as relevant to the results they show in the gut.

Minor comments

General to all figures. Please include statistical significance (*) or p-value in the cases where differences were statistically significant to make it easy to follow. If clock gene expression graphs are going to be compared, please use the same scale.

- Figure 1- In H scales are different and it's hard to compare the different levels. Also, in panel D please indicate if the dashed line is delimitating the intestine. In Figure 1 D -G I'm concerned that only one time point was measured. Could be that the clocks in pupa and larvae are present but the phase is different? Did the authors analyse any other time point?

- Figure 2- In Figure 2A scales are also different between graphs please use the same to help interpret the data, in the same panel, clk levels after 4 days look quite low, is hard to see the oscillation, please discuss in more detail why this could be happening. Why do PER and TIM reporters show different peaks of activity? In figure 2B use the same scale. TIM reporter has a peak at ZT0 and a minimum at ZT12 but PER reporter is delayed. Can you elaborate on this? In Figure S2A is histone=DAPI? Why B is the only one at ZT23?

- Figure 3. Please relocate panel D close to the other panels and the scale at the right bottom to make comparisons easier.

- Figure S4. When the authors said "We therefore used the higher-expressed clock genes as a 179 readout of clock development" they included some of the genes that did not change like dbt, vri,

and *cwo*, in this analysis the ones that clearly change are *Pdp1* and *Tim*. Please discuss why they decided to use the others too.

- Figure 5. In panel A GFP is not very clear, is there any other picture where the intensity is a bit higher?
- Figure S6A. When comparing ad libitum and starving flies, the size of the gut is affected by starvation but also the activity of the reporter, showing more activity when flies are in starvation mode, can authors elaborate on this.
- In line 316 change “fig.” for “figure”

Reviewer #3 (Remarks to the Author):

In this manuscript, Karpowicz and colleagues investigated when and how the circadian clock is established in the developing intestine of *Drosophila*. They utilized specific clock activity reporters, RT-qPCR analysis, and single cell RNA-sequencing analysis to study the emergence and regulation of the circadian clock during intestinal development. The authors observed that the circadian clock emerged in the adult intestine and established a daily rhythm three days after birth. They found increased circadian clock activities in ISCs and ECs but not EEs during intestinal maturation. The emergence of the clock after birth was regulated by ecdysone and Bursicon hormone signaling pathways. The authors also demonstrated that circadian clock gene expressions were repressed during ISC-to-EC differentiation. They further showed that photoperiod and feeding synchronized the maturation of the circadian clock during intestinal development.

While this study provides valuable insights into the emergence and regulation of the circadian clock during *Drosophila* intestine development, there are some areas that require further clarification and mechanistic insights. Here are specific comments:

Major:

1. In Figure 2B, the authors showed that the daily clock rhythm was not established until three days after eclosion. However, the changes in some genes were not as remarkable as described in the main text. It would be nice for the authors to provide alternative methods or evidence to support the establishment of the clock, such as demonstrating the establishment of rhythmic feeding (if it occurs) that is not observed until three days post eclosion.
2. The authors observed that the rhythmic clock is not active in EEs and in cells undergoing EB-to-EC differentiation. It would be helpful for the authors to explain how this happens. Do these cells have altered expression of hormone receptors, resulting in the loss of clock gene expression?

3. If the clock genes are restored in EEs and differentiating ECs, what would happen? This would provide insights into whether the temporal loss (for differentiating ECs) or permanent loss (for EEs) of the clock has any functional benefits.

4. In the cell lineage tracing experiments shown in Figure 5 and Figure S5, ISCs, EBs, and EEs are all small diploid cells in the lineage, which may not be reliably distinguished without co-staining with cell type-specific markers such as *DI*, *NRE-lacZ*, and *Pros*, respectively for ISCs, EBs and EEs.

5. The authors examined photoperiod and feeding as means to synchronize the circadian clock in the intestine, and the results were exciting and informative. However, it would be nice to include a negative control, such as fasted *cry01* mutant flies, to provide a comparison.

Minor:

1. As the process of ISC-EB-EC differentiation has been well-characterized in this study, the authors might want to consider examining the process of ISC-EEP-EE by extracting all progenitor cells and EEs from their single-cell data and performing dimension reduction analysis. Possibly the EEP population can be distinguished by candidate markers, such as *phyl* and *scute*. It would be interesting to understand when and how the clock genes disappear as ISCs differentiate into EEs.

2. The authors mentioned *Sox21a* as positively regulating ISCs, but two additional papers have shown that *Sox21a* negatively regulates ISC proliferation but positively regulates EB differentiation (Chen et al., eLife 2016; Zhai et al., Nat Comm 2015).

3. Line 279 mentions the "transient disappearance of circadian clock activity during lineage commitment of ISC to ECs in the adult intestine (Figure 4F)," but there is no Figure 4F.

Reviewer #1 (Remarks to the Author):

In this manuscript Parasram and colleagues examine the timecourse of appearance of clock function in the *Drosophila* intestine. Using a variety of different approaches they show that clock genes are not expressed in the intestine until the emergence of the adult fly, and that a robust circadian rhythm of clock gene expression does not appear until day 4 of adult life. Furthermore, they show that a mature clock first appears in the intestinal stem cells. Interestingly, they find that there is a transient loss of clock gene expression as cells transition between discrete fates. Finally, they show that the combination of daily photoperiod and feeding cycles together synchronize the timing of intestinal circadian transcription cycles during clock maturation.

The work reported here provides a careful description of the timecourse of gene expression in different cell types as the intestine develops from the larval to the adult stage. And, because the key clock genes are not expressed until adult emergence, it is clear from their analyses that there is no functional clock in the intestine until adulthood. Beyond this, however, this reviewer is not convinced about the exact timecourse of emergence of clock function in this tissue.

Specifically:

1- The results shown in Fig. 2A do not preclude the possibility that there is a functional clock by day 1, but that the clocks of different cells are not synchronized, leading to a lower overall rhythmicity. To address this question the authors would need to follow individual cells, and determine when a daily rhythm emerges. For this the tissue could be cultured and the intensity of their clock reporters in individual cells followed over time. (If I understood the legend correctly, in the timecourses shown in Figs. 1B and 1C each intestine contributed a single measurement, which I assume is a measure of average fluorescence intensity over an area of the intestine)

This live imaging experiment is an excellent idea and would certainly resolve these issues. However, our reporters use GFP and the blue light required for reporter imaging will reset the clock in culture (fly cells detect photoperiod light autonomously through Cry). To address the reviewer's question to the best of our ability, we have performed a characterization of the *Clock^{TIM}* reporter in adult ECs only spanning days 1-4 upon eclosion (Figure S3C). These reveal patterns similar to those observed in the whole intestine (initial rhythms become more stable/consolidated over this timeframe). If individual cells were not synchronized, resulting in a lack of rhythmicity, we would expect arrhythmicity at the early (day 1) stages, and heterogeneity in clock reporter expression in an individual intestine at any time point sampled (with some cells being bright and others darker). However, we see to a large extent uniform GFP fluorescence in individual guts (there is some variability between individuals as shown in our graphs, but less so in the cells in the same tissue), suggesting that individual cells are not at least anti-phasic relative to each other (Figure S2C, S3C). Nonetheless, the reviewer raises a good point regarding the strength of our conclusions in light of the new EC data we provide. The text has been updated to clarify this point (page 6): *"In each individual midgut, we did not notice major differences between cells which would indicate that the clocks in individual cells are at drastically different phases. To further test the synchrony between individual cells, we quantified EC fluorescence and noted that, although weaker rhythms are present at day 1, robust EC-specific rhythms are consolidated on day 4 (Figure S3C)."*

We have also updated the text to clarify the fluorescence measurements, as follows (page 6): *"To test this, we imaged the Clock^{PER} and Clock^{TIM} reporters and measured fluorescence intensity*

in the whole intestine following pupation every 3-hours over the first four days of adulthood (Figure 2B-C, S3A-B)."

The reviewer is correct that individual cells in the early stages of adulthood may show rhythms and thus we have revised some of our statements below (see response #2). However, our results still support the consolidation of synchronous rhythms in the intestine over the first 1-3 days after eclosion.

2- The statement (legend to Fig. 2) that "Clk/cyc transcriptional activity is initially stochastic and shows mature rhythms only by day 4 for Clock^{PER} with a peak around ZT0 and trough around ZT12." is not convincingly supported by the data. By eye, it looks like Fig. 1C shows a daily rhythm starting on day 1 with a daily minima at around ZT12. Regardless of what is seen by eye, it is really not possible to assert that rhythmicity does not emerge until day 4 absent a quantitative frequency analysis. Such a statement would also require having at least an extra day's worth of data, so as to have 2 days of robust rhythmicity (days 4-5) to contrast with the situation on days 1-2, for instance. I was also confused by the statistics provided (in Supplemental information), which for Figure 2B and 2C indicate the values of an ANOVA analysis. An ANOVA analysis only tells us that there are statistically significant differences among the values for different times, but this does not inform us as to whether there are daily rhythmicities and when such rhythmicity becomes significant.

In order to address these concerns, an additional two days (adult post-eclosion day 5 and 6), were tested and have been added to both of the Clock reporter analyses (Figure 2B-C). We used circacomp (R package) to perform additional cosinor analysis for day 1 and day 4 (Figure 2B-C), to compare these rhythmicity in early vs late adult intestines as recommended by the reviewer. The One-Way ANOVA was found to be significant in Clock^{PER} reporter and Clock^{TIM} on both day 1 and day 4, but the rhythmic p-value for Clock^{PER} at day 1 was not significant indicating Per transcription is not rhythmic at day 1. We also included cosinor analysis for days 1-2 and days 4-5 (shown in Figure S3B), although we note that, in these two-day results, that the gut is developing between day 1 and 2 so combining the tissue at two different states should be interpreted carefully. The strong peak and trough with Clock^{TIM} generates a curve that gradually synchronizes to the environment from day 1-4, whereas Clock^{PER} switches between arrhythmic and rhythmic until the rhythm stabilizes on day 4-6. Finally, we have examined Per protein nuclear localization and find that it is only robustly present and localized as expected at day 4 post-eclosion compared to day 1 (new data shown in Fig 2E). Overall, these suggest lower circadian clock rhythmicity at earlier (day 1) compared to later (day 4). The legend to figure 2 has been updated accordingly as follows (page 36): "... (B) Clk/cyc transcriptional activity in the Clock^{PER} reporter shows daily rhythms are established robustly on day 4 with a peak around ZT0 and trough around ZT12. Cosinor fit analysis shows arrhythmic activity on Day 1 and 24-hour rhythms on Day 4. (C) In the Clock^{TIM} reporter rhythms are noted earlier, from day 1, that phase shift to match those of Clock^{PER} by day 4."

The legend for Figure S3B reads (page 11, supplemental figures): "... (B) Cosinor analyses for days 1-2 compared with 4-5 for Clock^{PER} and Clock^{TIM} show slight shifts in the peak time during early clock activity. We interpret these with caution, however, since the intestine undergoes developmental maturation and may be dissimilar in the differentiated cell population on day 1 compared with day 2."

However, the reviewer is right in that we cannot exclude rhythms are present in the early intestine (day 1). To clarify our interpretation we have revised our findings to indicate that the rhythms

earlier are simply not robust circadian rhythms (page 6): “*When the adult ecloses, Clk/cyc transcriptional activity is present, but its rhythms are not robust 24-hour circadian oscillations until day 4 when these consolidate to exhibit a maximum in the morning (ZT21-ZT0) and minimum in the evening (ZT9-ZT12) characteristic of the Drosophila intestine [28].*”

Our final sentence in this section of the results reports our conclusions to take this point into account (page 7): “*Taken together, our results show that circadian clock development in the intestine has three distinct phases: (1) embryogenesis to late pupa when the circadian clock circuit is absent, (2) between adult day 1 to 3 when Clk/cyc initiate clock rhythmicity but not yet fully synchronized to the environment, and (3) day 4 when the circadian clock is synchronized and rhythmic in the mature intestine.*” We have also adjusted text in the Discussion (shown highlighted) to clarify this important point (page 27): “*Indeed, we find that clock gene expression is present in the early adult intestine but is not robustly synchronized and rhythmic until several days later (Figure 1-2).*” We hope that these convey the observation that rhythms may be present early but become more synchronous and robust later in adulthood.

3- Finally, the emergence of circadian function would also require records to be obtained under dark:dark (DD) conditions. The only such record is shown in Fig. S3, for which, incidentally, it isn't clear when the measurements were taken relative to the start of DD (ideally, measurements should be taken starting 24-48h after the beginning of DD, in case there is some residual rhythmicity). (As an aside, the legend to this figure states that “...full statistics are shown in Supplemental Information.”, yet this figure is part of Supplemental information and no statistics are shown in the “Statistics” table.)

To address this question, we collected samples for an additional 24-hours (CT24-48) under constant darkness to check for a free-running rhythm. We have updated the figure to show DD conditions (CT0-48) beginning on adult day 4 for *Clock^{TIM}* (now shown in Figure 2D). We apologize for the omission: the missing statistics for the DD experiment have also been added to the supplemental information (in the statistics chart for Figure 2D). The text has been updated to clarify the experimental conditions (page 7): “*Accordingly, Clock^{TIM} intestines were examined when flies were shifted to constant darkness (DD) prior to ZT0 on day 4 and tracked for 48-h (Figure 2D). The rhythms are similar to those under LD conditions, with a maximum in the morning (CT21-0) and minimum in the evening (CT12), demonstrating that Clk/cyc activity is free-running at day 4.*”

The figure legend has also been updated (page 36): “*(D) Clk/cyc activity when shifted to constant darkness prior to ZT0 on Day 4 shows free-running rhythms over the first two days in constant darkness. The vertical line separates the first and second days in constant darkness.*”

4- Many of the data have to do with changes associated with the maturation of the intestine rather than specifically that of circadian clock function (e.g., Fig. 4 and parts of Fig. 5). In addition, and as detailed above, the “circadian” part of the results is not very conclusive. For these reasons I would suggest changing the emphasis of the study and focusing on the maturation of the *Drosophila* intestine in general rather than more narrowly on the appearance of circadian clock function in this tissue.

The scRNAseq data (formerly figures 4-5) and the related screens show important changes related to maturation of the intestine and we use this analysis to identify factors that influence circadian clock development. Our study was very much focused on the circadian gene expression and development of the rhythmic oscillator from the conception to completion of the study. We believe we have discovered and characterized important aspects of circadian biology overall. We thank this reviewer for their valuable advice and believe that with the additional experiments

requested in comments #1-3, as well as additional Per protein analysis shown in Fig 2E, the circadian analysis in this paper is much stronger now.

5- The methods section needs additional details, in particular with respect to the measurements of fluorescence intensity.

The methods section has been updated to provide clarification for the fluorescence measurements, and a reference to a detailed description of this method that we published last year has been added (Parasram *et al.* 2022) (page 22): “*Fluorescence intensity measurements are ratios of GFP:DAPI taken for the whole midgut, unless otherwise specified, as previously described [111].*”

Minor comments:

- “phosphatase” should be deleted from line 106 “... the phosphatase dbt, a kinase that regulates...”.

The sentence was updated accordingly.

- The differences in color are not evident in the symbols shown in Figure 3 (especially purple vs. blue). Please make more pronounced.

The colours in the tSNE plots (Figure 3B) have been changed.

Reviewer #2 (Remarks to the Author):

In the manuscript titled “The Emergence of Circadian Timekeeping in the Intestine”, Parasram and colleagues characterized in detail the emergence and development of the circadian clock in the intestine of *Drosophila melanogaster*. In this work, researchers explored how different circadian genes are expressed during the developmental stages and in which intestinal cells these are more relevant. Using specific clock reporters, researchers showed changes in the activity and rhythmicity of TIM and PER in ISCs and their progeny cells, EB and EC. Importantly they concluded circadian rhythmicity and cell differentiation are exclusive processes, and the clock needs to be repressed in early differentiated cells (EB) to be restored in fully differentiated EC. Moreover, they demonstrated that expression of several genes started early in development (early pupa), but rhythmicity is only established in the adult after 4 days, highlighting the complexity of the circadian regulation in the gut. Interestingly feeding and photoperiod have to some extent compensatory effects, upon the lack of one or another, in influencing the establishment of circadian rhythmicity in the gut.

In summary, the manuscript demonstrates a novel approach to understanding circadian regulation during organ and tissue development and these findings contribute significantly to the existing body of knowledge in the field. This manuscript addresses a very important biological question, it is well-written and organized, making it easy to follow, ideas are clear, and figures and interpretations are well presented. Overall, data presented supports the main conclusions.

Major comments

- Clock reporter activity in ISCs is not clear from the images presented. For instance, in Figure S2F, while nuclear reporter activity is clearly observed in the ECs, it is not present in ISC/EB and therefore the claim of the authors regarding this is not supported by this evidence.

To show the Clock reporter activity in ISC/EBs, we replaced the images of reporter activity shown in Figure S2G (formerly S2F), and added an independent reporter line to further improve these (Figure S2G). In addition, we also replaced the images in Figure 6E-G with a third reporter line that better demonstrates the ISC/EB signal.

- There is a lack of information on the changes in the localization of PER at different time points. Authors should include a time course of cell specific staining of PER and/or TIM in the gut to substantiate the reporter information across intestinal development and cell types. For example, include a time course showing changes in localization of PER and/or TIM antibody comparing larvae, pupa, immature adult and mature adult gut 4 looking at the different cell types in the gut.

To test the changes in PER localization we used a GFP-PER fusion protein (*PER-AID-eGFP*) at ZT0 and ZT12 (now shown in Figure 2E). We have previously shown that ZT0 and ZT12 times correspond to adult nuclear and cytoplasmic localization of PER protein, respectively (Karpowicz, Zhang et al. 2013), but unfortunately we do not have any more of the Per or Tim antibodies left. Using the PER-AID-eGFP we were nonetheless able to complete the analysis requested by the reviewer. PER-AID-eGFP (per protein) expression is not present until the immature adult (adult day 1) consistent with our previous results. In the immature gut we notice ZT0 shows staining present in the nucleus and cytoplasm and at ZT12 we still see some cells with the same staining. This indicates that protein expression is not shuttling as expected in the mature adult, where Per protein is highly nuclear at ZT0 and not at ZT12. The text has been updated

accordingly (page 7): “In a functioning circadian system, clock proteins shuttle between the nucleus and cytoplasm [43]. We tested a GFP reporter, *per-AID-eGFP* [38] at two timepoints ZT0 and ZT12 that represent nuclear and cytoplasmic localization of *per*, respectively [27]. Larva and pupa do not show any expression of *per* consistent with our qPCR and reporter analysis; the immature adult (day 1) shows signal at ZT0 that remains at ZT12, suggesting that the protein shuttling is not yet established, while the mature adult (day 4) shows strong nuclear signal at ZT0 but not at ZT12 (Figure 2E).” The figure legend was also updated (page 36): “(E) Protein expression of *PER* (*PER-AID-eGFP*) at ZT0 and ZT12 shows that there is no *per* protein expression in larva or pupa, but *PER* is expressed in day 1 or day 4 adults with clear nuclear (ZT0) and cytoplasmic (ZT12) staining established by day 4.”

- The weakest point in this manuscript refers to the functional data on molecular regulators of clock gene expression based on the scRNAseq data (Figure 4). Is any of the data presented in Figure 4D statistically significant? What is the rationale behind knocking down these various genes in ECs and ISCs/EB, when the focus, as per the authors' statement and the transcriptional information presented in Figure 4B refers to the latter cell type? Furthermore, what is the evidence leading to Bursicon signaling (*rk*) as a possible player here? No previous evidence has been presented on expression or functional role of *rk* in ISCs or ECs and therefore the rationale for these experiments is unclear. The data in this part of the manuscript needs to be re-evaluated and improved/strengthened. I suggest choosing a couple of the strongest candidates and analyse their function in reporter expression over time in ISCs, coupled with immunostaining with core clock components (e.g. *PER*, *TIM*)

We agree with the reviewer that the data in Figure 4 needed to be clarified. We have redone this entire section and reevaluated our findings. In order to identify potential regulators of clock activity we first identified ISC-specific changes in the scRNAseq data (Figure 4). We could not analyze changes in ECs using scRNA-seq data since they are differentiating during this time (note the shift and additional clusters that appear from pupa to adult) whereas the ISC/EB population is more stable. However, if the factors turning on the clock are general regulatory mechanisms, we would expect that knockdowns in either ISC/EBs and ECs (as they differentiate – from EBs which are also clock-disrupted) would affect circadian clock activity in the intestine. We therefore initially considered ISC/EB knockdown but then compared to EC knockdown to compare the effect of hormone signaling in both cell types. To clarify this approach, we separated the functional genetic screening (Figure 5) from the scRNAseq analysis (Figure 4). We focused on candidates identified in Figure 4 and removed the gene, Bursicon, as suggested by the reviewer. For the regulators of clock expression (Figure 5), we now provide statistical significance in the colour coding of the violin plots, with groups significantly different compared to the control (blue) presented first in red and the non-significant groups in grey. These graphs have also been updated to include a more comprehensive set of nuclear receptors, including components of ecdysone and juvenile hormone signaling that are known to participate in adult maturation. To strengthen this analysis, as suggested by the reviewer, we then focused on some of our stronger candidates from both the scRNA-seq and the RNAi screen (Figure 4F-G), and performed overexpressions. The line that we used for the overexpression experiment contains an ISC/EB specific mCherry shown in the accompanying images. The text has also been updated to connect the two approaches (page 12): “To test whether these and other hormone nuclear receptors regulate clock development, we performed a screen by depleting genes in a cell-specific manner. Using our scRNA-seq analysis as a guide, we depleted components of hormone signalling in ISC/EBs (*esgTS*, 17 genes) using RNAi in *Clock*/*TIM* reporters during pupation (Fig 5A, Table S5). We reasoned that the loss of positive regulators of clock differentiation would delay *Clk/cyc*

activity, while negative regulators would hasten it. We found that (as predicted by SCENIC) *Clk/cyc* activity was reduced by *Hr78*, *Hnf4*, and *gce* knockdown, however, others such as *Met*, and *ftz-f1* did not (Figure 5A). Several other nuclear receptors were identified that also decrease *Clk/cyc* activity, but none were found that show an increase. To further test the requirement for hormonal signaling in activating clock gene expression, we overexpressed a subset of nuclear receptors identified in both the SCENIC analysis (Figure 4B) and tested by RNAi (Figure 5A). None of these were sufficient to accelerate *Clk/cyc* activity in pupal stages, suggesting that these pathways are required for circadian maturation but alone are not sufficient to prematurely drive clock emergence (Figure 5B). However, after eclosion *Clk/cyc* activity is significantly higher when *Hnf4* or *Hr78* are overexpressed, suggesting an important role in regulating intestinal clock development (Figure 5B). Since common regulatory pathways might affect subsequent clock activity in differentiating ECs as well, we retested EC-specific knockdowns of the same genes tested in ISC/EBs. Consistent with ISC/EBs, components of *Hnf4* signalling in ECs decreased *Clk/cyc* activity on the first day after eclosion, compared to the control (Figure 5C)."

Our overall finding is that certain factors, such as *Hnf4*, play a role in the development of the intestinal clock that is corroborated by scRNA-seq data analysis, knockdown, and overexpression. The Abstract, Discussion, and the Figure Legends have been updated accordingly to reflect these additional experiments and reorganization.

- Is the microbiome is playing a role in establishing the rhythmicity of the intestine during development and in the adult stages analysed?

Unlike the mammalian intestine, the *Drosophila* intestine's microbiome does not appear to show circadian rhythmicity, recently characterized in Zhang et al., 2023 (PMID: 36689661). Our experiments where flies were not provided food upon eclosion (which would affect their microbiome) also suggest it may not play a role (Fig 7A). However, as we did not test this explicitly, it is possible the microbiome could play a role in developing the clock and/or the tissue, thus we have updated the Discussion to include this possibility (page 17): "*In addition, environmental factors such as the microbiome might play a role in maturing both the tissue and its circadian rhythms, although it was recently reported that in the adult midgut the microbiome does not appear to show rhythmicity*[98]."

- Is there any role of the central clock in the establishment of intestinal clock during development?

This is an interesting question, although we did not test cell-specific disruption of the central clock, we did test the full body circadian clock mutant (*cyc⁰*) in which we overexpressed *cyc* in only *esg+* cells (ISC/EBs) (Figure S2I). In this case, *Clk/cyc* reporter activity emerges in the ISC/EBs in the intestine, which suggests that the circadian clock in ISCs can develop independently of other clocks in the body including the central clock. While this does not exclude the possibility that the central clock modulates the clocks in the gut, it does show that it is not absolutely required.

- Are all clocks in ISC synchronized or do you see different phases among them? This information could be obtained from the analysis of the ISC molecular signature in the scRNAseq. For example, if there were sub clusters of ISCs that express different levels or components of circadian genes.

We have previously found that the phase of the ISC clocks matches the rhythm between cells in this population and other cells in the tissue, overall suggesting that all the cells are indeed synchronized throughout the epithelium (Parasram, Bernardon et al. 2018). In the current dataset,

within the ISC population we do not see significant variation in either clock reporter activity or gene expression in the scRNAseq data, that would indicate separate sub clusters of ISCs in different phases of clock gene timing, or states where there are clock-active versus clock-dead ISC populations. Thus, our conclusions are that ISCs are generally synchronized.

- Consistent with the data presented here, Figure S5D, *pdp1*, is not essential for central clock expression and oscillations but for regulating circadian outputs, such as locomotion (Benito et al., 2007). Have the authors identified any function of *pdp1* in gut development or rhythmic behaviours (e.g. regeneration)? The mentioned prior evidence and associated publication should also be acknowledged by the authors as relevant to the results they show in the gut.

To address this issue, we tested a *Pdp1* mutant to see if this had an effect on the timing of the clock reporters during pupation. We did not find that *Pdp1* affects *Clk/cyc* activity in the developing intestine. We also attempted to overexpress *Pdp1* using *esg^{TS}* and *Myo1A^{TS}* in the midgut but due to survival effects were not able to get sufficient flies to complete this experiment. We have added the mutant data and the citation has been acknowledged (page 12): “*Our SCENIC analysis also suggests that Pdp1 may be a factor, however, Pdp1 loss (Pdp13135 mutant [75]) does not affect initiation of Clk/cyc activity (Figure S5A). This suggests that Pdp1 is not required for clock development, consistent with a role for Pdp1 in regulating clock outputs without being necessary for Clk expression [76].*”

We discuss the possible role of *Pdp1* in further detail in the Discussion (highlighted in yellow).

Minor comments

General to all figures. Please include statistical significance (*) or p-value in the cases where differences were statistically significant to make it easy to follow. If clock gene expression graphs are going to be compared, please use the same scale.

We have included * or p-values where space permitted, and similarly we have maintained the same scale to simplify comparison. In some cases (scatterplot overlay with line graphs), we report the statistical significance in the figure legends to avoid cluttering the figure. All statistical results are provided in a supplementary table as well.

• Figure 1- In H scales are different and it's hard to compare the different levels. Also, in panel D please indicate if the dashed line is delimitating the intestine. In Figure 1 D -G I'm concerned that only one time point was measured. Could be that the clocks in pupa and larvae are present but the phase is different? Did the authors analyse any other time point?

In Figure 1H, the different scales reflect different expression levels of the genes relative to the control, *Gapdh1*. While normalization to a peak could standardize the axis values, we found that this obscures important changes in expression levels that may bear biological relevance (i.e. *Vri* is 10x higher than *Clk* so would obscure it if the scale was exactly the same).

For panel D, we updated the figure legend with this clarification as requested (page 34): “...*(D) Images of larva, pupa, and adult show that Clock^{PER} is expressed only in adult intestines (Myo1A>mCherry). The intestine is outlined in a dashed line.*”

To address the reviewer's question about the clock phasing at different developmental stages in Fig 1D-G, we retested the clock reporters at different timepoints during the day (for example, ZT0/ZT12 corresponding to peak and trough of these) from larva to adult as well as additional

timepoints (Figure S1C-D, S2D). We found no Clk/cyc activity at any time of day until day 1 (after eclosion). We have updated the text with these additional results (page 5): “*We also verified that the clock is not present in larva and pupa, in a circadian phase opposite to the adults, by checking expression at different timepoints (Figure S2D).*”

- Figure 2- In Figure 2A scales are also different between graphs please use the same to help interpret the data, in the same panel, clk levels after 4 days look quite low, is hard to see the oscillation, please discuss in more detail why this could be happening. Why do PER and TIM reporters show different peaks of activity? In figure 2B use the same scale. TIM reporter has a peak at ZT0 and a minimum at ZT12 but PER reporter is delayed. Can you elaborate on this? In Figure S2A is histone=DAPI? Why B is the only one at ZT23?

Similar to Fig 1, we applied a different scale to the expression of clock genes relative to Gapdh. This is simply due to the expression level of these, for instance Cry is nearly 100x fold lower than Vri, so using the Cry scale would obscure any rhythms present in the Vri gene’s expression. As mentioned by the reviewer, the expression of *Clk* on Day 4 is not clearly rhythmic. It is possible that it is a reflection of the delay in the development of *Pdp1* and *vri* cycling which shows a strong change from day 1 to day 4 and regulates *Clk* transcription. The text in the Discussion has been updated accordingly (page 17): “*For instance, since Pdp1 and vri maintain rhythmic Clk expression [20], we note that their developmental maturation over days 1-4 may explain the initially low amplitude rhythms in Clk (Figure 2A).*”

The expression of Per and Tim, and their respective reporters is as slightly different phases as noted by the reviewer. This is consistent with our previous studies (Karpowicz, Zhang et al. 2013, Parasram, Bernardon et al. 2018), possibly because the promoters of these genes contains different elements that alter the exact timing of their expression. For Figure S2A, the staining is indeed histone not DAPI, this was one of the initial experiments performed (and coincidentally used the last of our PER antibody), we apologize for the inconsistency but do not think that it hinders the interpretation of this data since both histone and DAPI mark nuclei.

For Figure S2B, we noticed that in the original paper using the Cry-GFP (Agrawal et al., 2017) that samples are rapidly degraded in the presence of light. Due to the nature of these experiments we could not subject the flies to 3-days of DD however, we took this light sensitivity into consideration and found that the cry-GFP signal is best observed if stained without exposure to light, ZT23 (1-hour prior to lights-on), and the samples were kept in the dark throughout staining (manipulated using red light). The text in the Methods has been updated accordingly to explain (page 21): “*Cry-GFP was stained 1-hour prior to lights on (ZT23) to avoid light-induced degradation of cry [39].*”

- Figure 3. Please relocate panel D close to the other panels and the scale at the right bottom to make comparisons easier.

Panel D has been adjusted accordingly.

- Figure S4. When the authors said “We therefore used the higher-expressed clock genes as a readout of clock development” they included some of the genes that did not change like *dbt*, *vri*, and *cwo*, in this analysis the ones that clearly change are *Pdp1* and *Tim*. Please discuss why they decided to use the others too.

The tables in Figure S4 include the preliminary survey of clock gene expression in the scRNA-seq data, therefore we included the core clock genes (*per*, *tim*, *Clk*, *cyc*), the secondary loop (*Pdp1*, *vri*), and additional regulators (*dbt*, *cwo*). After considering this data, we were able to narrow our focus on *Pdp1*, *vri*, *tim* (which were also found in our qPCR data, Figure 1H) and a gene that is expressed in all samples (*cwo*), for further analysis due to their robust expression. The most dramatic difference is present in *Pdp1* and *Tim*, but we include additional genes as a more complete demonstration of changes to the clock during development. These genes were chosen based on a combination of their changes during development, and their overall gene expression level in these different assays.

- Figure 5. In panel A GFP is not very clear, is there any other picture where the intensity is a bit higher?

For Figure 5 (now Figure 6E), the experiments were repeated and better images were added to the manuscript.

- Figure S6A. When comparing ad libitum and starving flies, the size of the gut is affected by starvation but also the activity of the reporter, showing more activity when flies are in starvation mode, can authors elaborate on this.

We note overall that during the first few days of adulthood reporter expression is strongest, and this is the case after refeeding following starvation as shown in S6A. We have also noted this during regeneration (Parasram et al., 2018) where we also show that signaling pathways regulating proliferation (Hippo, Wnt) increase reporter activity. We speculate that Clk/cyc activity may increase during periods of regenerative growth but we are not sure at this point in time.

- In line 316 change “fig.” for “figure”

The text has been updated.

Reviewer #3 (Remarks to the Author):

In this manuscript, Karpowicz and colleagues investigated when and how the circadian clock is established in the developing intestine of *Drosophila*. They utilized specific clock activity reporters, RT-qPCR analysis, and single cell RNA-sequencing analysis to study the emergence and regulation of the circadian clock during intestinal development. The authors observed that the circadian clock emerged in the adult intestine and established a daily rhythm three days after birth. They found increased circadian clock activities in ISCs and ECs but not EEs during intestinal maturation. The emergence of the clock after birth was regulated by ecdysone and Bursicon hormone signaling pathways. The authors also demonstrated that circadian clock gene expressions were repressed during ISC-to-EC differentiation. They further showed that photoperiod and feeding synchronized the maturation of the circadian clock during intestinal development.

While this study provides valuable insights into the emergence and regulation of the circadian clock during *Drosophila* intestine development, there are some areas that require further clarification and mechanistic insights. Here are specific comments:

Major:

1. In Figure 2B, the authors showed that the daily clock rhythm was not established until three days after eclosion. However, the changes in some genes were not as remarkable as described in the main text. It would be nice for the authors to provide alternative methods or evidence to support the establishment of the clock, such as demonstrating the establishment of rhythmic feeding (if it occurs) that is not observed until three days post eclosion.

To address the reviewer's question about the gene expression changes in Figure 2B, we further tested the changes in gene transcription previously reported (new data spans Fig 2B-C) using circacomp (R package) to perform cosinor analysis comparing day 1 and day 4 Per and Tim reporter rhythms. The *Clock^{TIM}* reporter generates a curve that gradually synchronizes to the environment from days 1-4, whereas *Clock^{PER}* switches between arrhythmic and rhythmic until the rhythm stabilizes on day 4-6. The legend to figure 2 has been updated accordingly as follows (page 36): "... . (B) *Clk/cyc* transcriptional activity in the *ClockPER* reporter shows daily rhythms are established robustly on day 4 with a peak around ZT0 and trough around ZT12. Cosinor fit analysis (right graph) shows arrhythmic activity on Day 1 and 24-hour rhythms on Day 4. (C) In the *ClockTIM* reporter rhythms show a similar trend but are noted earlier, from day 1, that phase shift to match those of *ClockPER* by day 4."

These data provide quantitative analysis to support our observation that gene changes are different from day 1 compared to day 4. In order to provide further evidence to support the establishment of the clock in the intestine during development, we have added the characterization of Per protein using *PER-AID-eGFP* (Figure 2E) and tested more developmental times using this and the other clock reporters to see if anti-phasic rhythms or asynchronous rhythms are present at the cellular level. These new data support the delayed emergence of a circadian clock in the gut.

As mentioned in the Introduction, circadian behaviours are indeed present during *Drosophila* larval stages, and the central clock pacemaker that controls eclosion is established very early in the 1st instar larva (Sehgal, Price et al. 1992). Our study focussed only on the intestine, as an

example of a peripheral tissue where the ontogeny of the circadian clock was not yet established. We would predict that larval feeding is arrhythmic (they seem to feed continuously), but adult feeding is known to have daily rhythms (peak food consumption is early morning) (Xu, Zheng et al. 2008, Karpowicz, Zhang et al. 2013). Characterizing additional behavioural rhythms, including feeding, would be quite interesting to test during the different stages of *Drosophila* development. However, to test these behaviours we would need to build new systems/assays to accurately test larva since they appear to feed almost continuously. Pupa, of course, are immobile so would not elicit any of these. We feel this additional behavioural analysis falls outside the scope of the current manuscript given that the central pacemaker is established (larval stage) when the gut circadian pacemaker is not.

2. The authors observed that the rhythmic clock is not active in EEs and in cells undergoing EB-to-EC differentiation. It would be helpful for the authors to explain how this happens. Do these cells have altered expression of hormone receptors, resulting in the loss of clock gene expression?

To further examine the hormone receptor changes during differentiation, we used our scRNA-seq data to examine the changes in hormone receptor expression (Figure S5F). We focused on some of the candidates from our SCENIC data that show a similar pattern to the clock genes (high in ISCs, low in EBs, high in ECs). Several of these pathways indeed show a similar pattern to what we observed during the ISC lineage for clock genes. This suggests that nuclear receptors may indeed regulate clock activity during differentiation, and the text has been updated to include these new results (page 13): “*Since nuclear receptors play a role in activating Clk/cyc during development, we tested the expression of hormone signaling components in ISC lineages. Hnf4, Eip75B, and gce also show high expression in ISCs, low in EBs, returning in ECs (Figure S5C). This suggests that nuclear receptors are correlated with clock activity during adult ISC differentiation.*”

3. If the clock genes are restored in EEs and differentiating ECs, what would happen? This would provide insights into whether the temporal loss (for differentiating ECs) or permanent loss (for EEs) of the clock has any functional benefits.

This is a terrific experiment that would test the implications of our findings. However, it would be very difficult to restore clock genes in EEs and differentiating ECs, the former because most clock components are absent, and the latter because the transient stages of differentiation only last <2 days (~15h for EB formation plus ~23h for differentiation into an EC (Tang, Qin et al. 2021)). In a stable population such as EEs, this experiment would require ectopically expressing multiple clock genes (at least the Clk/cyc + Per/Tim components and the kinases that are required for their oscillatory activity). The level of overexpression would also have to be controlled so that several circadian cycles persist in these cells which do not have these genes present. We feel that, although compelling, this challenging experiment is beyond the scope of the current manuscript, and indeed requires knowledge of which components to express and control of how they are expressed to be informative. We note in our new dataset in Fig 5 that overexpression of specific hormone signaling components does not prematurely activate the clock, suggesting that multiple pathways are involved.

4. In the cell lineage tracing experiments shown in Figure 5 and Figure S5, ISCs, EBs, and EEs are all small diploid cells in the lineage, which may not be reliably distinguished without co-staining with cell type-specific markers such as D1, NRE-lacZ, and Pros, respectively for ISCs, EBs and EEs.

Due to the clock reporters using the GFP channel, we were not able to perform simultaneous imaging of the CD2 Flp/out clones (unfortunately, anti-CD2/DI/pros are all mouse antibodies and thus staining is not compatible with the CD2 Flp/out clones). We agree with the reviewer that this is a limitation of the lineage tracing experiment, and it is important distinguish between the small diploid cells (ISCs/EBs/EEs). To address this concern, we have included improved imaging of the *Clock^{PER}* and *Clock^{TIM}* reporters using *Su(H)GBE>mCherry* to mark EBs specifically, and *esg-Gal4;Su(H)GBE-Gal80>mCherry* to mark ISCs specifically (now shown in Figure 6E-G, and S2G). Our previous work has tested these reporters in the EE cell population extensively and found that EEs are clock-negative cells (Parasram, Bernardon et al. 2018). Together with the scRNA-seq we report in this study, we are confident of the assessment of clock gene expression in the different diploid cell types present in this tissue.

5. The authors examined photoperiod and feeding as means to synchronize the circadian clock in the intestine, and the results were exciting and informative. However, it would be nice to include a negative control, such as fasted *cry01* mutant flies, to provide a comparison.

To address the reviewer's question, we have added a fasted control for the *cry⁰¹* flies (Figure 7C). We initially tried using fasting (water only) but the *cry⁰¹* flies do not survive long-term fasting, dying after ~2 days. We therefore subjected *cry⁰¹* flies to alternate days of feeding and fasting (water only) to disrupt any rhythms due to feeding behaviour. The resulting Clk/cyc rhythm is arrhythmic (without food restriction the *cry⁰¹* cannot synchronize). These experiments have been added to the paper (page 14): "*To further test this, we restricted feeding in cry01 flies post-pupation to the morning only (ZT0-6), evening only (ZT5-11), or subjected them to alternate days of feeding and starvation to disrupt feeding rhythmicity. We predicted that the timing of food consumption would alter the maxima and minima of intestinal circadian rhythms. Indeed, under these conditions the rhythms of cry01 intestines were shifted to peak around 6h after feeding, or rendered non-circadian when disrupted (no clear 24-hour period) (Figure 7C).*"

Minor:

1. As the process of ISC-EB-EC differentiation has been well-characterized in this study, the authors might want to consider examining the process of ISC-EEP-EE by extracting all progenitor cells and EEs from their single-cell data and performing dimension reduction analysis. Possibly the EEP population can be distinguished by candidate markers, such as *Phyl* and *Scute*. It would be interesting to understand when and how the clock genes disappear as ISCs differentiate into EEs.

As suggested, we tried to use our scRNA-seq data to perform a similar lineage analysis for the EEs. Unfortunately, we were not able to detect the markers such as *Phyl* and *Scute* in our dataset and were not able to reconstruct a specific cell lineage for EEs. This may be due to a drop-out effect of the *Phyl* and *Scute* markers in our scRNA-seq data. If we use the same well-characterized cells that form the intermediate state (between ISC and EC – shown in Fig 6) we note a clock disappearance starts in the EB state (as shown in Figure 6B), however, without the ability to reexamine the process specifically in the EE lineage we did not add this reconstruction to the present study. However, the supplemental table 4 has been updated to include the markers of each of the progenitor clusters to clarify the cell types we have been able to identify.

2. The authors mentioned *Sox21a* as positively regulating ISCs, but two additional papers have shown that *Sox21a* negatively regulates ISC proliferation but positively regulates EB differentiation (Chen et al., eLife 2016; Zhai et al., Nat Comm 2015).

The text has been corrected and the additional citations have been added (page 10):
*“Identification of cell-specific transcription factors revealed enrichment of *klu* and *Sox21a* in ISCs, known to regulate ISC differentiation/function [48, 54, 55, 56], the Notch target *E(spl)* in ISC/EBs, known to drive ISC differentiation [57, 58], *nub/Pdm1* in ECs, a well-known marker of this cell type [23, 24], *tap* in EEs [59], known to promote EE cell fate [60] (Figure 4A).”*

3. Line 279 mentions the "transient disappearance of circadian clock activity during lineage commitment of ISC to ECs in the adult intestine (Figure 4F)," but there is no Figure 4F.

The figure number (Figure 6H) has been updated in the text.

References (above)

- Karpowicz, P., Y. Zhang, J. B. Hogenesch, P. Emery and N. Perrimon (2013). "The circadian clock gates the intestinal stem cell regenerative state." Cell reports **3**(4): 996-1004.
- Parasram, K., N. Bernardon, M. Hammoud, H. Chang, L. He, N. Perrimon and P. Karpowicz (2018). "Intestinal Stem Cells Exhibit Conditional Circadian Clock Function." Stem cell reports **11**(5): 1287-1301.
- Sehgal, A., J. Price and M. W. Young (1992). "Ontogeny of a biological clock in *Drosophila melanogaster*." Proceedings of the National Academy of Sciences **89**(4): 1423-1427.
- Tang, R., P. Qin, X. Liu, S. Wu, R. Yao, G. Cai, J. Gao, Y. Wu and Z. Guo (2021). "Intravital imaging strategy FlyVAB reveals the dependence of *Drosophila* enteroblast differentiation on the local physiology." Commun Biol **4**(1): 1223.
- Xu, K., X. Zheng and A. Sehgal (2008). "Regulation of feeding and metabolism by neuronal and peripheral clocks in *Drosophila*." Cell metabolism **8**(4): 289-300.

REVIEWERS' COMMENTS

Reviewer #1 (Remarks to the Author):

The authors have made a valiant effort to revise their manuscript and to respond to the reviewers' comments and suggestions. I have no further comments.

Reviewer #2 (Remarks to the Author):

The revised version of this manuscript significantly addresses most of my major comments. Before publication, I would however suggest further clarification on the following points:

The expression of the Clock reporter in ISCs/EBs remains unclear. What does the data in Figure S2G mean? There are two sets of data, one showing no reporter activity in either pupal or adult ISCs/EBs and the other does show reporter activity.

I don't understand the data provided in Figure S2I. If *cyc* is over expressed in ISCs/EBs in the whole mutant animal, would one not expect to restore the reporter activity in these cells? I supposed the same would be the case in EC if the gene is over expressed in those cells?

Figure 5B: Over-expression of *Hnf4* and *Hr78* in ISC/EBs leads to clock activity increase in ECs. Why is that?

Reviewer #3 (Remarks to the Author):

The majority of my concerns have been addressed by the authors. However, there are still a few minor issues that need attention:

The curve for day 1 is missing in Fig. 2B.

There are several citation mismatches. For instance, in line 199, Fig. S4E should be Fig. S4F.

Please see our point-by-point response to the reviewer's comments/questions below.

Reviewer #1 (Remarks to the Author):

The authors have made a valiant effort to revise their manuscript and to respond to the reviewers' comments and suggestions. I have no further comments.

Reviewer #2 (Remarks to the Author):

The revised version of this manuscript significantly addresses most of my major comments. Before publication, I would however suggest further clarification on the following points:

The expression of the Clock reporter in ISCs/EBs remains unclear. What does the data in Figure S2G mean? There are two sets of data, one showing no reporter activity in either pupal or adult ISCs/EBs and the other does show reporter activity.

Figure S2G (now Figure S3G) shows ISC/EB expression of the *Clock^{TIM}* reporter using two different lines that show differences in the strength of the GFP signal. *Clock^{TIM}* (on chromosome III) expression, which is quantified in the violin plots, has a weaker GFP signal, than *Clock^{TIM}* (II). To demonstrate unambiguously the ISC/EB signal, we included images from both reporters. To clarify this point, we have adjusted the Figure legend as follows:

“(G) Clk/cyc activity in ISC/EBs (marked by *mCherry* expression in *esg+* cells, representative cells are outlined) is similar in late pupa and immature adults, Mann-Whitney test p-values are shown on the graph. Two reporters are shown, the quantification corresponds to *Clock^{TIM}* (III) but due to its weaker expression in ISC/EBs, we have included an image of *Clock^{TIM}* (II) which shows a similar pattern but stronger GFP expression in ISC/EBs.”

I don't understand the data provided in Figure S2I. If *cyc* is over expressed in ISCs/EBs in the whole mutant animal, would one not expect to restore the reporter activity in these cells? I supposed the same would be the case in EC if the gene is over expressed in those cells?

In Figure S2I (now Supplementary Figure 3I) we restore *cyc* expression only in the ISC/EBs in the whole mutant animal. The ISC/EBs are marked with *mCherry*, and we see the reporter restored in the ISC/EBs. We have adjusted the text and image panels to clarify this point and indicate cells with reporter expression.

“(I) Representative images of *Clock^{TIM}* intestines showing that the Clk/cyc activity that is present in the controls, is absent in the *cyc⁰¹* mutant, and restored strongly in the ISC/EBs when *cyc* is

overexpressed (*esg>mCherry, cyc*). Arrowheads indicate mCherry+ ISC/EBs. Scale bar 10 μ m. Full statistics are shown in Supplemental Information.”

Yes, although we have not done the EC overexpression experiment, we would expect this to be the same case in ECs if the gene is overexpressed in those cells.

Figure 5B: Over-expression of Hnf4 and Hr78 in ISC/EBs leads to clock activity increase in ECs. Why is that?

The ISC/EB overexpression of nuclear receptors does affect EC reporter signal. We expect that the earlier Clk/cyc activity in ISC/EBs (precursors of ECs) allows the EC clock to develop earlier as well, since the ISCs would give rise to these clock positive cells earlier. The text has been updated to include this observation.

“Of note, EC reporter expression is also increased when these genes are overexpressed in ISCs, possibly due to the higher clock activity in their precursors.”

Reviewer #3 (Remarks to the Author):

The majority of my concerns have been addressed by the authors. However, there are still a few minor issues that need attention:

The curve for day 1 is missing in Fig. 2B.

The curve for day 1 in Figure 2B is missing because the *Clock^{PER}* reporter is arrhythmic at day 1, therefore it does not fit a cosinor curve which could not be added to the graph. The figure legend has been updated with this clarification.

“Cosinor fit analysis (right graph) shows arrhythmic activity on Day 1 (no cosinor curve can be fitted to the data) and 24-hour rhythms on Day 4.”

There are several citation mismatches. For instance, in line 199, Fig. S4E should be Fig. S4F.

We apologize for these mismatches; we have verified that the citations are matching in this final version.